# Provably Robust Conformal Prediction with Improved Efficiency

**Ge Yan**
CSE, UCSD
geyan@ucsd.edu

**Yaniv Romano**
ECE, Technion
yromano@technion.ac.il

**Tsui-Wei Weng**
HDSI, UCSD
lweng@ucsd.edu

## Abstract

Conformal prediction is a powerful tool to generate uncertainty sets with guaranteed coverage using any predictive model, under the assumption that the training and test data are i.i.d.. Recently, it has been shown that adversarial examples are able to manipulate conformal methods to construct prediction sets with invalid coverage rates, as the i.i.d. assumption is violated. To address this issue, a recent work, Randomized Smoothed Conformal Prediction (RSCP), was first proposed to certify the robustness of conformal prediction methods to adversarial noise. However, RSCP has two major limitations: (i) its robustness guarantee is flawed when used in practice and (ii) it tends to produce large uncertainty sets. To address these limitations, we first propose a novel framework called `RSCP+` to provide provable robustness guarantee in evaluation, which fixes the issues in the original RSCP method. Next, we propose two novel methods, Post-Training Transformation (PTT) and Robust Conformal Training (RCT), to effectively reduce prediction set size with little computation overhead. Experimental results in CIFAR10, CIFAR100, and ImageNet suggest the baseline method only yields trivial predictions including full label set, while our methods could boost the efficiency by up to $4.36\times$, $5.46\times$, and $16.9\times$ respectively and provide practical robustness guarantee.

## 1 Introduction

Conformal prediction (Lei & Wasserman, 2014; Papadopoulos et al., 2002; Vovk et al., 2005) has been a powerful tool to quantify prediction uncertainties of modern machine learning models. For classification tasks, given a test input $x_{n+1}$, it could generate a prediction set $C(x_{n+1})$ with coverage guarantee:

$$\mathbb{P}[y_{n+1} \in C(x_{n+1})] \geq 1 - \alpha, \tag{1}$$

where $y_{n+1}$ is the ground truth label and $1 - \alpha$ is user-specified target coverage. This property is desirable in safety-critical applications like autonomous vehicles and clinical applications. In general, it is common to set the coverage probability $1 - \alpha$ to be high, e.g. 90% or 95%, as we would like the ground truth label to be contained in the prediction set with high probability. It is also desired to have the smallest possible prediction sets $C(x_{n+1})$ as they are more informative. In this paper, we use the term "efficiency" to compare conformal prediction methods: we say a conformal prediction method is more efficient if the size of the prediction set is smaller.

Despite the power of conformal prediction, recent work (Gendler et al., 2021) showed that conformal prediction is unfortunately prone to adversarial examples – that is, the coverage guarantee in Eq. (1) may not hold anymore because adversarial perturbation on test data breaks the i.i.d. assumption and thus the prediction set constructed by vanilla conformal prediction becomes invalid. To solve this problem, Gendler et al. (2021) proposes a new technique, named Randomized Smoothed Conformal Prediction (RSCP), which is able to construct new prediction sets $C_\epsilon(\tilde{x}_{n+1})$ that is robust to adversarial examples:

$$\mathbb{P}[y_{n+1} \in C_\epsilon(\tilde{x}_{n+1})] \geq 1 - \alpha, \tag{2}$$

where $\tilde{x}_{n+1}$ denotes a perturbed example that satisfies $\|\tilde{x}_{n+1} - x_{n+1}\|_2 \leq \epsilon$ and $\epsilon > 0$ is the perturbation magnitude. The key idea of RSCP is to modify the vanilla conformal prediction procedure with randomized smoothing (Cohen et al., 2019; Duchi et al., 2012; Salman et al., 2019) so that the impact of adversarial perturbation could be bounded and compensated.

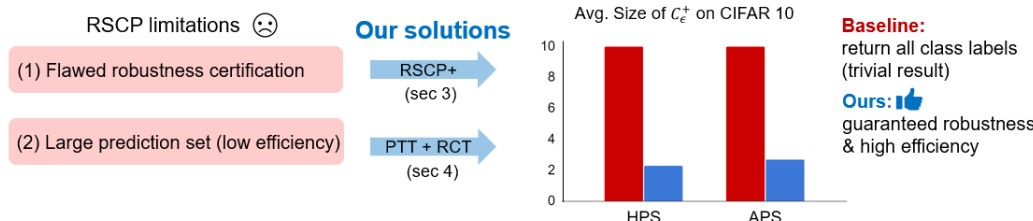

Figure 1: An overview of this work: We address two limitations of RSCP (Gendler et al., 2021) by proposing `RSCP+` (Sec. 3) & PTT + RCT (Sec. 4), which enables the first *provable* and *efficient* robust conformal prediction. As we show in the experiments in Sec. 5, our proposed method could provide useful robust prediction sets information while the baseline failed.

However, RSCP has two major limitations: (1) *the robustness guarantee of RSCP is flawed*: RSCP introduces randomized smoothing to provide robustness guarantee. Unfortunately, the derived guarantee is invalid when Monte Carlo sampling is used for randomized smoothing, which is how randomized smoothing is implemented in practice (Cohen et al., 2019). Therefore, their robustness certification is invalid, despite empirically working well. (2) *RSCP has low efficiency*: The average size of prediction sets of RSCP is much larger than the vanilla conformal prediction, as shown in our experiments (Fig. D.1).

In this paper, we will address these two limitations of RSCP to allow *efficient* and *provably robust* conformal prediction by proposing a new theoretical framework `RSCP+` in Sec. 3 to guarantee robustness, along with two new methods (PTT & RCT) in Sec. 4 to effectively decrease the prediction set size. We summarize our contributions below:

1. We first identify the major issue of RSCP in robustness certification and address this issue by proposing a new theoretical framework called `RSCP+`. The main difference between `RSCP+` and RSCP is that our `RSCP+` uses the Monte Carlo estimator directly as the base score for RSCP, and amends the flaw of RSCP with simple modification on the original pipeline. To our best knowledge, `RSCP+` is the first method to provide *practical certified robustness* for conformal prediction.

2. We further propose two methods to improve the efficiency of `RSCP+`: a scalable, training-free method called PTT and a general robust conformal training framework called RCT. Empirical results suggest PTT and RCT are necessary for providing guaranteed robust prediction sets.

3. We conduct extensive experiments on CIFAR10, CIFAR100 and ImageNet with `RSCP+`, PTT and RCT. Results show that without our method the baseline only gives trivial predictions, which are uninformative and useless. In contrast, our methods provide practical robustness certification and boost the efficiency of the baseline by up to $4.36\times$ on CIFAR10, $5.46\times$ on CIFAR100, and $16.9\times$ on ImageNet.

## 2 BACKGROUND AND RELATED WORKS

### 2.1 CONFORMAL PREDICTION

Suppose $D = \{(x_i, y_i)\}_{i=1}^n$ is an i.i.d. dataset, where $x_i \in \mathbb{R}^p$ denotes the features of $i$th sample and $y_i \in [K] := \{1, \ldots, K\}$ denotes its label. Conformal prediction method divides $D$ into two parts: a training set $D_{\text{train}} = \{(x_i, y_i)\}_{i=1}^m$ and a calibration set $D_{\text{cal}} = D \setminus D_{\text{train}}$. The training set $D_{\text{train}}$ is utilized to train a classifier function $\hat{\pi}(x) : \mathbb{R}^p \to [0, 1]^K$. Given classifier $\hat{\pi}$, a non-conformity score function $S(x, y) : \mathbb{R}^p \times [K] \to \mathbb{R}$ is defined for each class $y$ based on classifier's prediction $\hat{\pi}(x)$. Next, the calibration set $D_{\text{cal}}$ is utilized to calculate threshold $\tau$, which is the $(1 - \alpha)(1 + 1/|D_{\text{cal}}|)$ empirical quantile of calibration scores $\{S(x, y)\}_{(x,y) \in D_{\text{cal}}}$. Given a test sample $x_{n+1}$, conformal prediction construct a prediction set $C(x_{n+1}; \tau)$ as:

$$C(x_{n+1}; \tau) = \{k \in [K] \mid S(x_{n+1}, k) \leq \tau\}, \tag{3}$$

where

$$\tau = Q_{1-\alpha}(\{S(x,y)\}_{(x,y)\in D_{\text{cal}}}) \tag{4}$$

and $Q_p(D_{\text{cal}})$ denote the $p(1 + 1/|D_{\text{cal}}|)$-th empirical quantile of the calibration scores. In the remainder of the paper, we may omit the parameter $\tau$ and write the prediction set simply as $C(x)$ when the context is clear. Conformal prediction ensures the coverage guarantee in Eq. (1) by showing that the score corresponding to the ground truth label is bounded by $\tau$ with probability $1 - \alpha$, i.e. $\mathbb{P}(S(x_{n+1}, y_{n+1}) \le \tau) \ge 1 - \alpha$.

Note that the above conformal prediction pipeline works for any non-conformity score $S(x,y)$, but the statistical efficiency of conformal prediction is affected by the choice of non-conformity score. Common non-conformity scores include HPS (Lei et al., 2013; Sadinle et al., 2019) and APS (Romano et al., 2020):

$$S_{\text{HPS}}(x,y) = 1 - \hat{\pi}_y(x), \ S_{\text{APS}}(x,y) = \sum_{y'\in[K]} \hat{\pi}_{y'}(x)\mathbb{1}_{\{\hat{\pi}_{y'}(x) > \hat{\pi}_y(x)\}} + \hat{\pi}_y(x)\cdot u, \tag{5}$$

where $u$ is a random variable sampled from a uniform distribution over $[0, 1]$.

## 2.2 RANDOMIZED SMOOTHED CONFORMAL PREDICTION

To ensure the coverage guarantee still holds under adversarial perturbation, Gendler et al. (2021) proposed *Randomized Smoothed Conformal Prediction (RSCP)*, which defines a new non-conformity score $\tilde{S}$ that can construct new prediction sets that are robust against adversarial attacks. The key idea of RSCP is to consider the worst-case scenario that $\tilde{S}$ may be affected by adversarial perturbations:

$$\tilde{S}(\tilde{x}_{n+1}, y) \le \tilde{S}(x_{n+1}, y) + M_\epsilon, \forall y \in [K], \tag{6}$$

where $x_{n+1}$ denotes the clean example, $\tilde{x}_{n+1}$ denotes the perturbed example that satisfies $\|\tilde{x}_{n+1} - x_{n+1}\|_2 \le \epsilon$ and $M_\epsilon$ is a non-negative constant. Eq. (6) indicates that the new non-conformity score $\tilde{S}$ on adversarial examples may be inflated, but fortunately the inflation can be bounded. Therefore, to ensure the guarantee in Eq. (2) is satisfied, the threshold $\tau$ in the new prediction set needs to be adjusted to $\tau_{\text{adj}}$ defined as $\tau_{\text{adj}} = \tau + M_\epsilon$ to compensate for potential adversarial perturbations, and then $C_\epsilon$ can be constructed as follows:

$$C_\epsilon(x; \tau_{\text{adj}}) = \{k \in [K] \mid \tilde{S}(x, k) \le \tau_{\text{adj}}\}, \tag{7}$$

where $x$ is any test example. From Eq. (6), the validity of $C_\epsilon$ could be verified by following derivation:

$$y_{n+1} \in C(x_{n+1}) \Rightarrow \tilde{S}(x_{n+1}, y_{n+1}) \le \tau \Rightarrow \tilde{S}(\tilde{x}_{n+1}, y_{n+1}) \le \tau_{\text{adj}} \Rightarrow y_{n+1} \in C_\epsilon(\tilde{x}_{n+1}). \tag{8}$$

Thus, the coverage guarantee in Eq. (2) is satisfied. To obtain a valid $M_\epsilon$, Gendler et al. (2021) proposed to leverage randomized smoothing (Cohen et al., 2019; Duchi et al., 2012) to construct $\tilde{S}$. Specifically, define

$$\tilde{S}(x,y) = \Phi^{-1}[S_{\text{RS}}(x,y)] \text{ and } S_{\text{RS}}(x,y) = \mathbb{E}_{\delta\sim\mathcal{N}(0,\sigma^2 I_p)} S(x+\delta, y), \tag{9}$$

where $\delta$ is a Gaussian random variable, $\sigma$ is the standard deviation of $\delta$ which controls the strength of smoothing, and $\Phi^{-1}(\cdot)$ is Gaussian inverse cdf. We call $S_{\text{RS}}(x,y)$ the randomized smoothed score from a base score $S(x,y)$, as $S_{\text{RS}}(x,y)$ is the smoothed version of $S(x,y)$ using Gaussian noise on the input $x$. Since $\Phi^{-1}$ is defined on the interval $[0, 1]$, the base score $S$ must satisfy $S(x,y) \in [0, 1]$. One nice property from randomized smoothing (Cohen et al., 2019) is that it guarantees that $\tilde{S}$ is Lipschitz continuous with Lipschitz constant $\frac{1}{\sigma}$, i.e. $\frac{|\tilde{S}(\tilde{x}_{n+1}, y_{n+1}) - \tilde{S}(x_{n+1}, y_{n+1})|}{\|\tilde{x}_{n+1} - x_{n+1}\|_2} \le \frac{1}{\sigma}$. Hence, we have

$$\|\tilde{x}_{n+1} - x_{n+1}\|_2 \le \epsilon \implies \tilde{S}(\tilde{x}_{n+1}, y_{n+1}) \le \tilde{S}(x_{n+1}, y_{n+1}) + \frac{\epsilon}{\sigma}, \tag{10}$$

which is exactly Eq. (6) with $M_\epsilon = \frac{\epsilon}{\sigma}$. Therefore, when using $\tilde{S}$ in conformal prediction, the threshold should be adjusted by:

$$\tau_{\text{adj}} = \tau + \frac{\epsilon}{\sigma}. \tag{11}$$

## 3 CHALLENGE 1: ROBUSTNESS GUARANTEE

In this section, we point out a flaw in the robustness certification of RSCP (Gendler et al., 2021) and propose a new scheme called RSCP+ to provide provable robustness guarantee in practice. As we discuss in Sec. 2.2, the key idea of RSCP is introducing a new conformity score $\tilde{S}$ that satisfies Eq. (10), which gives an upper bound to the impact of adversarial perturbation. However, in practice, $\tilde{S}$ is intractable due to expectation calculation in $S_{RS}$. A common practice in randomized smoothing literature is:

- **Step 1:** Approximate $S_{RS}$ by Monte Carlo estimator:

$$\hat{S}_{RS}(x, y) = \frac{1}{N_{MC}} \sum_{i=1}^{N_{MC}} S(x + \delta_i, y), \delta_i \sim \mathcal{N}(0, \sigma^2 I_p). \tag{12}$$

- **Step 2:** Bound the estimation error via some concentration inequality.

In RSCP, however, **Step 2** is missing, because bounding the error simultaneously on the calibration set is difficult, as discussed in Appendix A.1. We argue that the missing error bound makes the robustness guarantee of RSCP invalid in practice.

To address this issue, we propose an elegant and effective approach, RSCP+, to fill in the gap and provide the guarantee. In particular, the intrinsic difficulty in bounding Monte Carlo error inspires us to avoid the estimation. Thus, in RSCP+ we propose a new approach to incorporate the Monte Carlo estimator $\hat{S}_{RS}$ directly as the (non-)conformity score, which could be directly calculated, unlike $S_{RS}$. Here, one question may arise is: Can a randomized score (e.g. $\hat{S}_{RS}$) be applied in conformal prediction and maintain the coverage guarantee? The answer is yes: as we discuss in Appendix A.2, many classical (non-)conformity scores (e.g. APS (Romano et al., 2020)) are randomized scores, and the proofs for them are similar to the deterministic scores, as long as the i.i.d. property between calibration and test scores is preserved. Therefore, our $\hat{S}_{RS}$ is a legit (non-)conformity score.

The challenge of using $\hat{S}_{RS}$ is to derive an inequality similar to Eq. (10), i.e. connect $\hat{S}_{RS}(\tilde{x}_{n+1}, y)$ and $\hat{S}_{RS}(x_{n+1}, y)$ (the grey dotted line in Fig. 2), so that we can bound the impact from adversarial noises and compensate for it accordingly. To achieve this, we use $S_{RS}$ as a bridge (as shown in Fig. 2), and present the result in Theorem 1.

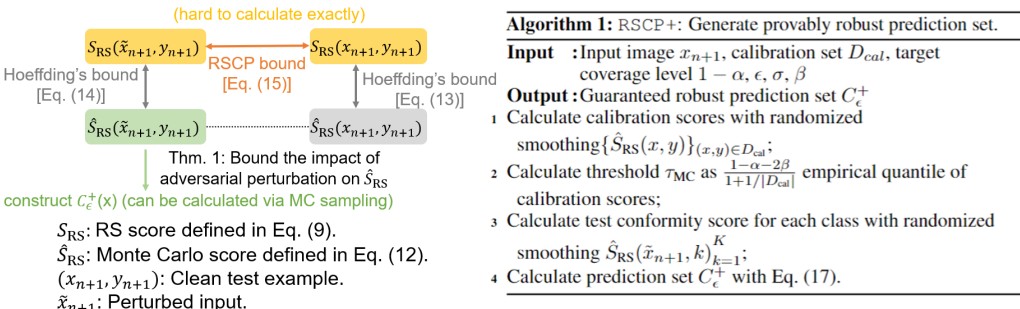

Figure 2: Diagram illustrating our RSCP+. (Left) (1) The dotted line shows our target: bound Monte-Carlo estimator score $\hat{S}_{RS}$ under perturbation; (2) The orange arrow denotes the bound of the randomized smoothed score $S_{RS}$ under perturbation, given by (Gendler et al., 2021); (3) The grey arrows denote Hoeffding's inequality connecting randomized smoothed score $S_{RS}$ and Monte Carlo estimator score $\hat{S}_{RS}$. The target (1) could be derived by (2) + (3). (Right) RSCP+ algorithm.

**Theorem 1.** *Let $(x_{n+1}, y_{n+1})$ be the clean test sample and $\tilde{x}_{n+1}$ be perturbed input data that satisfies $\|\tilde{x}_{n+1} - x_{n+1}\|_2 \leq \epsilon$. Then, with probability $1 - 2\beta$:*

$$\hat{S}_{RS}(\tilde{x}_{n+1}, y_{n+1}) - b_{Hoef}(\beta) \leq \Phi\left[\Phi^{-1}[\hat{S}_{RS}(x_{n+1}, y_{n+1}) + b_{Hoef}(\beta)] + \frac{\epsilon}{\sigma}\right],$$

*where $b_{Hoef}(\beta) = \sqrt{\frac{-\ln\beta}{2N_{MC}}}$, $N_{MC}$ is the number of Monte Carlo examples, $\Phi$ is standard Gaussian cdf, $\sigma$ is smoothing strength and $\hat{S}_{RS}$ is the Monte Carlo score defined in Eq. (12).*

*Proof of Theorem 1.* The main idea of the proof is connecting $\hat{S}_{\text{RS}}(x_{n+1}, y_{n+1})$ and $\hat{S}_{\text{RS}}(\tilde{x}_{n+1}, y_{n+1})$ via the corresponding $S_{\text{RS}}$, as shown in Fig. 2. By Hoeffding's inequality (See Appendix A.3 for further discussion), we have

$$S_{\text{RS}}(x_{n+1}, y_{n+1}) \leq \hat{S}_{\text{RS}}(x_{n+1}, y_{n+1}) + b_{\text{Hoef}}(\beta) \tag{13}$$

by Eq. (A.8) and

$$S_{\text{RS}}(\tilde{x}_{n+1}, y_{n+1}) \geq \hat{S}_{\text{RS}}(\tilde{x}_{n+1}, y_{n+1}) - b_{\text{Hoef}}(\beta) \tag{14}$$

by Eq. (A.9), both with probability $1 - \beta$. Meanwhile, by plugging in the definition of $\tilde{S}$, Eq. (10) is equivalent to

$$\Phi^{-1}[S_{\text{RS}}(\tilde{x}_{n+1}, y_{n+1})] \leq \Phi^{-1}[S_{\text{RS}}(x_{n+1}, y_{n+1})] + \frac{\epsilon}{\sigma}. \tag{15}$$

Combining the three inequalities above and applying union bound gives:

$$S_{\text{RS}}(\tilde{x}_{n+1}, y_{n+1}) \leq \Phi\left[\Phi^{-1}[S_{\text{RS}}(x_{n+1}, y_{n+1})] + \frac{\epsilon}{\sigma}\right]$$

$$\xrightarrow[\text{with prob. } 1-\beta]{\text{Eq. (13)}} S_{\text{RS}}(\tilde{x}_{n+1}, y_{n+1}) \leq \Phi\left[\Phi^{-1}[\hat{S}_{\text{RS}}(x_{n+1}, y_{n+1}) + b_{\text{Hoef}}] + \frac{\epsilon}{\sigma}\right] \tag{16}$$

$$\xrightarrow[\text{with prob. } 1-2\beta]{\text{Eq. (14)}} \hat{S}_{\text{RS}}(\tilde{x}_{n+1}, y_{n+1}) - b_{\text{Hoef}}(\beta) \leq \Phi\left[\Phi^{-1}[\hat{S}_{\text{RS}}(x_{n+1}, y_{n+1}) + b_{\text{Hoef}}(\beta)] + \frac{\epsilon}{\sigma}\right],$$

with probability $1 - 2\beta$, which proves Theorem 1. $\qquad\square$

**Remark.** *The bound in Theorem 1 could be further improved using Empirical Bernstein's inequality (Maurer & Pontil, 2009). We found in our experiments that the improvement is light on CIFAR10 and CIFAR100, but could be significant on ImageNet. For more discussion see Appendix A.3.3.*

With Theorem 1, we could construct the prediction set accordingly and derive the robustness guarantee in Corollary 2 in the following.

**Corollary 2.** *(Robustness guarantee for* `RSCP+`*) The* `RSCP+` *prediction set*

$$C_\epsilon^+(\tilde{x}_{n+1}; \tau_{MC}) = \left\{ k \in [K] \mid \hat{S}_{RS}(\tilde{x}_{n+1}, k) - b_{Hoef}(\beta) \leq \Phi\left[\Phi^{-1}[\tau_{MC} + b_{Hoef}(\beta)] + \frac{\epsilon}{\sigma}\right]\right\} \tag{17}$$

*satisfies robust coverage guarantee in Eq. (2), i.e.* $\mathbb{P}(y_{n+1} \in C_\epsilon^+(\tilde{x}_{n+1}; \tau_{MC})) \geq 1 - \alpha$*. Here, the threshold* $\tau_{MC}$ *is calculated according to Eq. (4) with* $S = \hat{S}_{RS}$ *and* $1 - \alpha$ *replaced by* $1 - \alpha + 2\beta$*, i.e.* $\tau_{MC} = Q_{1-\alpha+2\beta}(\{\hat{S}_{RS}(x, y)\}_{(x,y) \in D_{cal}})$*.*

*Proof of Corollary 2.* Since we have $\tau_{\text{MC}} = Q_{1-\alpha+2\beta}(\{\hat{S}_{\text{RS}}(x, y)\}_{(x,y) \in D_{\text{cal}}})$, conformal prediction guarantees coverage on clean examples:

$$\mathbb{P}[\hat{S}_{\text{RS}}(x_{n+1}, y_{n+1}) \leq \tau_{\text{MC}}] \geq 1 - \alpha + 2\beta. \tag{18}$$

Plug Eq. (18) into Eq. (16) in Theorem 1 and apply union bound, we get

$$\mathbb{P}\left\{\hat{S}_{\text{RS}}(\tilde{x}_{n+1}, y_{n+1}) - b_{\text{Hoef}}(\beta) \leq \Phi\left[\Phi^{-1}[\tau_{\text{MC}} + b_{\text{Hoef}}(\beta)] + \frac{\epsilon}{\sigma}\right]\right\} \geq 1 - \alpha. \tag{19}$$

$$\square$$

# 4 CHALLENGE 2: IMPROVING EFFICIENCY

So far, we have modified RSCP to `RSCP+` that can provide a certified guarantee in Sec. 3. However, there exists another challenge – directly applying `RSCP+` often leads to trivial prediction sets that give the entire label set, as shown in our experiment Tabs. 1 and 2. The reason is that RSCP is *conservative*: instead of giving an accurate coverage as vanilla CP, RSCP attains a higher coverage due to its threshold inflation (Eq. (11)), and thus gives a larger prediction set on both clean and perturbed data. We define *conservativeness* of RSCP as the increase in the average size of prediction sets after threshold inflation: see Appendix A.4 where we give a formal definition. Since `RSCP+` is modified from RSCP, it's expected to inherit the conservativeness, leading to trivial predictions. To address this challenge and make `RSCP+` useful, in this section, we propose to address this problem by modifying the base score $S$ with two new methods: Post Training Transformation (PTT) and Robust Conformal Training (RCT).

## 4.1 POST-TRAINING TRANSFORMATION (PTT)

**Intuition.** We first start with a quantitative analysis of the conservativeness by threshold inflation. As an approximation to the conservativeness, we measure the coverage gap between inflated coverage $1 - \alpha_{\text{adj}}$ and target coverage $1 - \alpha$:

$$\alpha_{\text{gap}} = (1 - \alpha_{\text{adj}}) - (1 - \alpha) = \alpha - \alpha_{\text{adj}}. \tag{20}$$

Next, we conduct a theoretical analysis on $\alpha_{\text{gap}}$. Let $\Phi_{\tilde{S}}(t)$ be the cdf of score $\tilde{S}(x,y)$, where $(x,y) \sim P_{xy}$. For simplicity, suppose $\Phi_{\tilde{S}}(t)$ is known. Recall that in conformal prediction, the threshold $\tau$ is the minimum value that satisfies the coverage condition:

$$\tau = \underset{t \in \mathbb{R}}{\arg\min} \left\{ \mathbb{P}_{(x,y) \sim P_{xy}}[\tilde{S}(x,y) \leq t] \geq (1 - \alpha). \right\} \tag{21}$$

Notice that $\mathbb{P}_{(x,y) \sim P_{xy}}[\tilde{S}(x,y) \leq t]$ is exactly $\Phi_{\tilde{S}}(t)$, we have:

$$\Phi_{\tilde{S}}(\tau) = 1 - \alpha. \tag{22}$$

Suppose the threshold is inflated as $\tau_{\text{adj}} = \tau + M_\epsilon$. Similarly, we could derive $1 - \alpha_{\text{adj}} = \Phi_{\tilde{S}}(\tau_{\text{adj}}) = \Phi_{\tilde{S}}(\tau + M_\epsilon)$ by Eq. (11). Now the coverage gap $\alpha_{\text{gap}}$ can be computed as:

$$\alpha_{\text{gap}} = \alpha - \alpha_{\text{adj}} = \Phi_{\tilde{S}}(\tau + M_\epsilon) - \Phi_{\tilde{S}}(\tau) \approx \Phi'_{\tilde{S}}(\tau) \cdot M_\epsilon \tag{23}$$

The last step is carried out by the linear approximation of $\Phi_{\tilde{S}}$: $g(x + z) - g(x) \approx g'(x) \cdot z$.

**Key idea.** Eq. (23) suggests that we could reduce $\alpha_{\text{gap}}$ by **reducing the slope** of $\Phi_{\tilde{S}}$ near the original threshold $\tau$, i.e. $\Phi'_S(\tau)$. This inspires us to the idea: can we perform a transformation on $\tilde{S}$ to reduce the slope while keeping the information in it? Directly applying transformation on $\tilde{S}$ is not a valid option because it would break the Lipschitz continuity of $\tilde{S}$ in Eq. (10): for example, applying a discontinuous function on $\tilde{S}$ may make it discontinuous. However, we could apply a transformation $\mathcal{Q}$ on the base score $S$, which modifies $\tilde{S}$ indirectly while preserving the continuity, as long as the transformed score, $\mathcal{Q} \circ S$, still lies in the interval $[0, 1]$. The next question is: how shall we design this transformation $\mathcal{Q}$? Here, we propose that the desired transformation $\mathcal{Q}$ should satisfy the following two conditions:

1. **(Slope reduction)** By applying $\mathcal{Q}$, we should reduce the slope $\Phi'_{\tilde{S}}(\tau)$, thus decrease the coverage gap $\alpha_{\text{gap}}$. Since we are operating on base score $S$, we approximate this condition by reducing the slope $\Phi'_S(\tau)$. We give a rigorous theoretical analysis of a synthetic dataset and an empirical study on real data to justify the effectiveness of this approximation in Appendices B.6 and B.7, respectively.

2. **(Monotonicity)** $\mathcal{Q}$ should be monotonically non-decreasing. It could be verified that under this condition, $(\mathcal{Q} \circ S)$ is equivalent to $S$ in vanilla CP (See our proof in Appendix B.5). Hence, the information in $S$ is kept after transformation $\mathcal{Q}$.

These two conditions ensure that transformation $\mathcal{Q}$ could alleviate the conservativeness of RSCP without losing the information in the original base score. With the above conditions in mind, we design a two-step transformation $\mathcal{Q}$ by composing **(I)** ranking and **(II)** Sigmoid transformation on base score $S$, denoted as $\mathcal{Q} = \mathcal{Q}_{\text{sig}} \circ \mathcal{Q}_{\text{rank}}$. We describe each transformation below.

**Transformation (I): ranking transformation $\mathcal{Q}_{\text{rank}}$.** The first problem we encounter is that we have no knowledge about the score distribution $\Phi_S$ in practice, which makes designing transformation difficult. To address this problem, we propose a simple data-driven approach called ranking transformation to turn the unknown distribution $\Phi_S$ into a uniform distribution. With this, we could design the following transformations on it and get the analytical form of the final transformed score distribution $\Phi_{\mathcal{Q} \circ S}$. For ranking transformation, we sample an i.i.d. holdout set $D_{\text{holdout}} = \{(x_i, y_i)\}_{i=1}^{N_{\text{holdout}}}$ from $P_{XY}$, which is disjoint with the calibration set $D_{\text{cal}}$. Next, scores $\{S(x,y)\}_{(x,y) \in D_{\text{holdout}}}$ are calculated on the holdout set and the transformation $\mathcal{Q}_{\text{rank}}$ is defined as:

$$\mathcal{Q}_{\text{rank}}(s) = \frac{r\left[s; \{S(x,y)\}_{(x,y) \in D_{\text{holdout}}}\right]}{|D_{\text{holdout}}|}.$$

Here, $r(x; H)$ denotes the rank of $x$ in set $H$, where ties are broken randomly. We want to emphasize that this rank is calculated on the holdout set $D_{\text{holdout}}$ for both calibration samples and test samples. We argue that the new score $\mathcal{Q}_{\text{rank}} \circ S$ is uniformly distributed, which is a well-known result in statistics(Kuchibhotla, 2020). See more discussion in Appendix B.3.

**Transformation (II): Sigmoid transformation $\mathcal{Q}_{\textbf{sig}}$.** After ranking transformation, we get a uniformly distributed score. The next goal is reducing $\Phi'_S(\tau)$. For this, we introduce Sigmoid transformation $\mathcal{Q}_{\text{sig}}$. In this step, a sigmoid function $\phi$ is applied on $S$:

$$\mathcal{Q}_{\text{sig}}(s) = \phi\left[(s - b)/T\right],$$

where $b, T$ are hyper-parameters controlling this transformation. Due to space constraint, we discuss more details of Sigmoid transformation in Appendix B.4, where we show that the distribution of transformed score $\Phi_{\mathcal{Q}_{\text{sig}} \circ \mathcal{Q}_{\text{rank}} \circ S}$ is the inverse of Sigmoid transformation $\mathcal{Q}_{\text{sig}}^{-1}$ (Eq. (B.2)), and by setting $b = 1 - \alpha$ and $T$ properly small, the Sigmoid transformation could reduce $\Phi'_S(\tau)$.

**Summary.** Combining ranking transformation and sigmoid transformation, we derive a new (non-)conformity score $S_{\text{PTT}}$:

$$S_{\text{PTT}}(x, y) = (\mathcal{Q}_{\text{sig}} \circ \mathcal{Q}_{\text{rank}} \circ S)(x, y). \tag{24}$$

It could be verified that $S_{\text{PTT}}(x, y) \in [0, 1]$ for any $S$ thanks to the sigmoid function, hence we could plug in $S \leftarrow S_{\text{PTT}}(x, y)$ into Eq. (9) as a base score. Additionally, $S_{\text{PTT}}(x, y)$ is monotonically non-decreasing, satisfying the monotonicity condition described at the beginning of this section. We provide a rigorous theoretical study on PTT over on a synthetic dataset in Appendix B.7. Additionally, we craft a case in Appendix B.8 where PTT may not improve the efficiency. Despite this theoretical possibility, we observe that PTT consistently improves over the baseline in experiments.

## 4.2 ROBUST CONFORMAL TRAINING (RCT)

While our proposed PTT provides a training-*free* approach to improve efficiency, there is another line of work (Einbinder et al., 2022b; Stutz et al., 2021) studying how to train a better base classifier for conformal prediction. However, these methods are designed for standard conformal prediction instead of *robust* conformal prediction considered in our paper. In this section, we introduce a training pipeline called RCT, which simulates the RSCP process in training to further improve the efficiency of robust conformal prediction.

**Conformal training.** Stutz et al. (2021) proposed a general framework to train a classifier for conformal prediction. It simulates conformal prediction in training by splitting the training batch $B$ into a calibration set $B_{\text{cal}}$ and a prediction set $B_{\text{pred}}$, then performing conformal prediction on them. The key idea is to use soft surrogate $\tau^{\text{soft}}$ and $c(x, y; \tau^{\text{soft}})$ to approximate the threshold $\tau$ and prediction set $C(x; \tau)$, making the pipeline differentiable: $\tau^{\text{soft}} = Q_{1-\alpha}^{\text{soft}}(\{S_\theta(x, y)\}_{(x,y) \in B_{\text{cal}}})$, where $Q_q^{\text{soft}}(H)$ denotes the $q(1 + \frac{1}{|H|})$-quantile of set $H$ derived by smooth sorting (Blondel et al., 2020; Cuturi et al., 2019), and $c(x, y; \tau^{\text{soft}}) = \phi\left[\frac{\tau^{\text{soft}} - S_\theta(x,y)}{T_{\text{train}}}\right]$, where $\phi(z) = 1/(1 + e^{-z})$ is the sigmoid function and temperature $T_{\text{train}}$ is a hyper-parameter. We introduce more details in Appendix C.1.

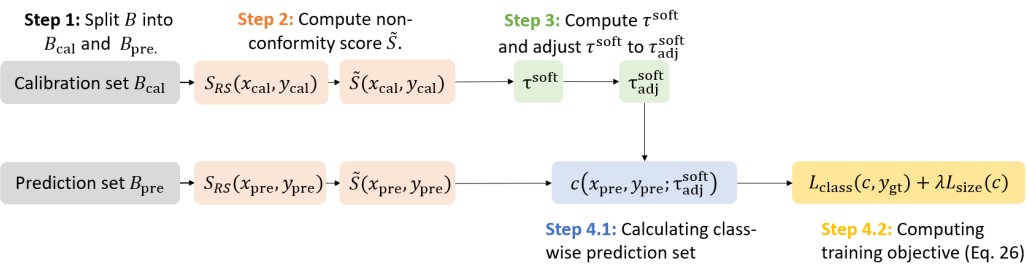

Figure 3: Pipeline of our proposed Robust Conformal Training (RCT) method.

**Incorporating RSCP into training.**   Inspired by Stutz et al. (2021), we propose to incorporate RSCP (Gendler et al., 2021) (and of course, `RSCP+` since the major steps are the same) into the training stage as shown in Fig. 3. We adopt soft threshold $\tau^{\text{soft}}$ and soft prediction $c(x, y; \tau^{\text{soft}})$ from Stutz et al. (2021), and add randomized smoothing $\tilde{S}$ and threshold adjustment $\tau_{\text{adj}}^{\text{soft}} = \tau^{\text{soft}} + \frac{\epsilon}{\sigma}$ to the pipeline as in RSCP. Next, we need to examine the differentiability of our pipeline. The threshold adjustment and Gaussian inverse cdf $\Phi^{-1}$ step in the calculation of $\tilde{S}$ is differentiable, but the gradient of $S_{\text{RS}} = \mathbb{E}_{\delta \sim \mathcal{N}(0, \sigma^2 I_p)} S(x + \delta, y)$ is difficult to evaluate, as the calculation of $S(x, y)$ involves a deep neural network and expectation. Luckily, several previous works (Salman et al., 2019; Zhai et al., 2020) have shown that the Monte-Carlo approximation works well in practice:

$$\nabla_\theta \mathbb{E}_{\delta \sim \mathcal{N}(0, \sigma^2 I_p)} S(x + \delta, y) \approx \frac{1}{N_{\text{train}}} \sum_{i=1}^{N_{\text{train}}} \nabla_\theta S(x + \delta_i, y). \tag{25}$$

With these approximations, the whole pipeline becomes differentiable and training could be performed by back-propagation. For the training objective, we can use the same loss function:

$$L(x, y_{\text{gt}}) = L_{\text{class}}(c(x, y; \tau^{\text{soft}}), y_{\text{gt}}) + \lambda L_{\text{size}}(c(x, y; \tau^{\text{soft}})), \tag{26}$$

where classification loss $L_{\text{class}}(c(x, y; \tau^{\text{soft}}), y_{\text{gt}}) = 1 - c(x, y_{\text{gt}}; \tau^{\text{soft}})$, size loss $L_{\text{size}}(c(x, y; \tau^{\text{soft}})) = \max(0, \sum_{y=1}^{K} c(x, y; \tau^{\text{soft}}) - \kappa)$, $y_{\text{gt}}$ denotes the ground truth label, $c(x, y; \tau^{\text{soft}})$ denotes the soft prediction introduced in Stutz et al. (2021), $\kappa$ is a hyper-parameter.

**Remark.** *Since the methods we proposed in Sec. 4 (PTT and RCT) are directly applied to base scores, they are orthogonal to the `RSCP+` we proposed in Sec. 3. That is to say, PTT and RCT not only work on `RSCP+` but also work on original RSCP as well. Nevertheless, we argue that `RSCP+` with PTT/RCT would be more desirable in practice since it provides **guaranteed robustness** which is the original purpose of provable robust conformal prediction. Hence, we will focus on this benchmark in the experiments section in the main text. However, we also provide experiment results on RSCP + PTT/RCT as an empirical robustness benchmark in Appendix D.2, which shows that our PTT and RCT are not limited to our `RSCP+` scheme.*

## 5   EXPERIMENTS

In this section, we evaluate our methods in Secs. 3 and 4. Experiments are conducted on CIFAR10, CIFAR100 (Krizhevsky et al., 2009) and ImageNet (Deng et al., 2009) and target coverage is set to $1 - \alpha = 0.9$. We choose perturbation magnitude $\epsilon = 0.125$ on CIFAR10 and CIFAR100 and $\epsilon = 0.25$ on ImageNet.

**Evaluation metrics and baseline.** We present the average size of prediction sets $C_\epsilon^+(x)$ as a key metric, since the robustness is guaranteed by our theoretical results for `RSCP+`(Corollary 2). For the baseline, we choose the vanilla method from Gendler et al. (2021), where HPS and APS are directly applied as the base score without any modifications.

**Model.** We choose ResNet-110 (He et al., 2016) for CIFAR10 and CIFAR100 and ResNet-50 (He et al., 2016) for ImageNet. The pre-trained weights are from Cohen et al. (2019) for CIFAR10 and ImageNet and from Gendler et al. (2021) for CIFAR100.

**Hyperparameters.** In `RSCP+`, we choose $\beta = 0.001$ and the number of Monte Carlo examples $N_{\text{MC}} = 256$. For PTT, we choose $b = 0.9$ and $T = 1/400$ and we discuss this choice in Appendix B.4. The size of holdout set $|D_{\text{holdout}}| = 500$. We discuss more experimental details in Appendix D.

### 5.1   RESULTS AND DISCUSSION

Tab. 1 and Tab. 2 compare the average size of prediction sets on all three datasets with our `RSCP+` benchmark. Specifically, the first row shows the baseline method using base scores in Gendler et al. (2021) directly equipped with our `RSCP+`. Note that the baseline method gives trivial prediction sets (the prediction set size = total number of class, which is totally uninformative) due to its conservativeness. Our methods successfully address this problem and provide a meaningful prediction set with robustness guarantee.

| Base score | HPS | | APS | |
|---|---|---|---|---|
| Method / Dataset | CIFAR10 | CIFAR100 | CIFAR10 | CIFAR100 |
| Baseline (Gendler et al., 2021) | 10 | 100 | 10 | 100 |
| PTT (**Ours**) | **2.294** | 26.06 | **2.685** | 21.96 |
| PTT+RCT (**Ours**) | **2.294** | **18.30** | 2.824 | **20.01** |
| Improvement over baseline: PTT | **4.36×** | **3.84×** | **3.72×** | **4.55×** |
| Improvement over baseline: PTT + RCT | **4.36×** | **5.46×** | **3.54×** | **5.00×** |

Table 1: **Average prediction set ($C_\epsilon^+(x)$) size of `RSCP+` on CIFAR10 and CIFAR100.** For CIFAR10 and CIFAR100, $\epsilon = 0.125$ and $\sigma = 0.25$. Following Gendler et al. (2021), we take $N_{\text{split}} = 50$ random splits between calibration set and test set and present the average results (Same for Tab. 2). We could see that the baseline method only gives trivial predictions containing the whole label set, while with PTT or PTT + RCT we can give informative and compact predictions.

| Method / Base score | HPS | APS |
|---|---|---|
| Baseline (Gendler et al., 2021) | 1000 | 1000 |
| PTT (**Ours**) | 1000 | 94.66 |
| PTT + Bernstein (**Ours**) | **59.12** | **70.87** |
| Improvement over baseline: PTT | - | **10.6×** |
| Improvement over baseline: PTT + Bernstein | **16.9×** | **14.1×** |

Table 2: **Average prediction set ($C_\epsilon^+(x)$) size of `RSCP+` on ImageNet.** For ImageNet, $\epsilon = 0.25$ and $\sigma = 0.5$. The ImageNet dataset is more challenging and our PTT only works for APS score, but we find by applying the improvement with Empirical Bernstein's bound (denoted as "PTT + Bernstein") we discussed in Appendix A.3.3, we could largely reduce the size of prediction sets.

| $N_{MC}$ | 256 | 512 | 1024 | 2048 | 4096 |
|---|---|---|---|---|---|
| Average size of prediction sets $C_\epsilon^+(x)$ | 2.294 | 2.094 | 1.954 | 1.867 | 1.816 |

Table 3: **Average size vs. Number of Monte Carlo samples $N_{MC}$.** The experiment is conducted on CIFAR10 dataset with PTT method. The base score is HPS. It could be seen that by increasing the number of Monte Carlo examples, we could further improve the efficiency of `RSCP+`, at the cost of higher computational expense.

**Why the baseline gives trivial results under `RSCP+`?** The key reason is conservativeness. RSCP is conservative compared to vanilla conformal prediction, and the challenging task of giving guaranteed robustness makes the situation worse. The result is that: without the boost of our PTT and RCT methods, the predictor is so conservative that it gives the whole label set to guarantee robustness, which is not the goal of users. This again justifies the necessity of our methods.

**Impact of number of Monte Carlo samples $N_{\text{MC}}$.** In Tab. 3, we study how the number of Monte Carlo samples ($N_{\text{MC}}$) influences the average size. It could be observed that the average size decreases as more Monte Carlo samples are taken. This is expected as more Monte Carlo samples reduce the error and provide a tighter bound in Eqs. (13) and (14). Therefore, a trade-off between prediction set size and computation cost needs to be considered in practice, since increasing $N_{\text{MC}}$ also significantly boosts the computation requirement.

## 6 CONCLUSION

This paper studies how to generate prediction sets that are robust to adversarial attacks. We point out that the previous method RSCP (Gendler et al., 2021) has two major limitations: flawed robustness certification and low efficiency. We propose a new theoretically sound framework called `RSCP+` which resolves the flaw in RSCP and provides a provable guarantee. We also propose a training-free and scalable method (PTT) and robust conformal training method (RCT) to significantly boost the efficiency of RSCP. We have conducted extensive experiments and the empirical results support our theoretical analysis. Experiments show that the baseline gives trivial prediction sets (all class labels), while our methods are able to provide meaningful prediction sets that boost the efficiency of the baseline by up to $4.36\times$ on CIFAR10, $5.46\times$ on CIFAR100, and $16.9\times$ on ImageNet.

## ACKNOWLEDGEMENT

This work is supported in part by National Science Foundation (NSF) awards CNS-1730158, ACI-1540112, ACI-1541349, OAC-1826967, OAC-2112167, CNS-2100237, CNS-2120019, the University of California Office of the President, and the University of California San Diego's California Institute for Telecommunications and Information Technology/Qualcomm Institute. Thanks to CENIC for the 100Gbps networks. G. Yan and T.-W. Weng are supported by National Science Foundation under Grant No. 2107189 and 2313105. Y. Romano was supported by the Israel Science Foundation (grant No. 729/21). Y. Romano also thanks the Career Advancement Fellowship, Technion, for providing research support.

## REPRODUCIBILITY STATEMENT

We provide the training details of RCT, hyperparameters, and other details of our experiments in Appendix D. The code is released on https://github.com/Trustworthy-ML-Lab/Provably-Robust-Conformal-Prediction.

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

APPENDIX

## A    FURTHER DISCUSSION ON RSCP+ IN SEC 3

### A.1    WHY BOUNDING MONTE CARLO ESTIMATION ERROR IS DIFFICULT FOR ORIGINAL RSCP?

As we discussed in the main text, in the calculation of the conformity score of RSCP, Monte-Carlo estimation for $S_{\text{RS}}$ is required which introduces estimation error. However, in the original work (Gendler et al., 2021), the error bound for this is not introduced. Below we discuss why this kind of bound is difficult to be applied in RSCP context.

The main difficulty lies in the threshold calculation step: denote $Q_p(H)$ the $p(1 + 1/|H|)$-empirical quantile of a set $H$ and recall that $\tau_{\text{adj}} = \tau + \frac{\epsilon}{\sigma} = Q_{1-\alpha}(\{\tilde{S}(x,y)\}_{(x,y)\in D_{\text{cal}}}) + \frac{\epsilon}{\sigma}$ depends on every example in the calibration set. This means that to derive a non-trivial bound for $\tau_{\text{adj}}$, we need to bound the error on the whole set *simultaneously*, which is difficult. For example, if there are 10000 samples in the calibration set and each example is bounded with a confidence level of 0.999, then the confidence level for threshold $\tau_{\text{adj}}$ would be $(0.999)^{10000} \approx 4.5 \times 10^{-5}$! To obtain a reasonable confidence level for $\tau_{\text{adj}}$, practitioners have to use a much smaller calibration set or a much higher confidence level, which may significantly hurt the performance.

### A.2    DOES THE COVERAGE GUARANTEE OF CONFORMAL PREDICTION HOLD FOR RANDOMIZED SCORES?

In the proof of Corollary 2, we utilize the coverage guarantee of conformal prediction on clean examples, i.e. Eq. (18). A natural question is, does the coverage guarantee of conformal prediction still hold when a randomized score (like $\hat{S}_{RS}$ we introduced) is applied?

**We argue the answer is yes.**    Randomized conformity scores have been used in multiple previous works, e.g. APS by Romano et al. (2020) and RAPS by Angelopoulos et al. (2020). The theory they presented is slightly different from our case, but the key idea is the same. Below, we present a proof for Eq. (18) under our settings. Similar results could be found in (Angelopoulos et al., 2020; Romano et al., 2020).

**Lemma A.1.** *Assume that $D_{cal} = \{(x_i, y_i)\}_{i=1}^n$ is an i.i.d. sampled calibration set, $(x_{n+1}, y_{n+1})$ is a test example which is independently sampled from the same distribution. The score of example $i$ is $S_i = \hat{S}_{RS}(x_i, y_i) = \frac{1}{N_{MC}} \sum_{j=1}^{N_{MC}} S(x_i + \delta_{ij}, y_i), \forall i = 1, 2, \cdots, n+1$, where all Monte-Carlo noises $\delta_{ij}$ are i.i.d. sampled from $\mathcal{N}(0, \sigma^2 I_p)$, independent from other variables. Then, $\{S_i\}_{i=1}^{n+1}$ are i.i.d. distributed.*

*Proof.* Note that the randomness of score $S_i$ comes from the data $(x_i, y_i)$ and the Monte-Carlo noise $\delta_{ij}$. From our assumption, $(x_i, y_i)$ are i.i.d. across the calibration set and test sample, and the noise $\delta_{ij}$ are also drawn i.i.d. independent of other variables. Since $S_i$ is a function of $(x_i, y_i)$ and $\{\delta_{ij}\}_{j=1}^{N_{\text{MC}}}$, this leads to the conclution that $\{S_i\}_{i=1}^{n+1}$ are i.i.d. distributed.    □

**Lemma A.2.** *Suppose $\{S_i\}_{i=1}^{n+1}$ are i.i.d. distributed random variables. $\tau_{MC}$ is the $q(1 + 1/n)$-th empirical quantile of $\{S_i\}_{i=1}^n$, i.e.*

$$\tau_{MC} = min_s \frac{|\{i \mid S_i \leq s\}|}{n} \geq q(1 + \frac{1}{n}).$$

*Then, $\mathbb{P}(S_{n+1} \leq \tau_{MC}) \geq q$.*

*Proof.* We define the following indicator variables:

$$I_i = \mathbf{1}_{|\{j|S_i > S_j\}| \geq \lceil (n+1)q \rceil}.$$

These variables have the following two properties:

1. $\{I_i\}_{i=1}^{n+1}$ are identically distributed. Thus, $\mathbb{E}I_1 = \mathbb{E}I_2 = \cdots = \mathbb{E}I_{n+1}$. This follows from the i.i.d. property of $\{S_i\}_{i=1}^{n+1}$ and symmetricity.

2. $\sum_{i=1}^{n+1} I_i \leq (n+1) - \lceil (n+1)q \rceil$. Suppose we order $S_i$ from the smallest to the largest: $S_{q_1} \leq S_{q_2} \cdots \leq S_{q_{n+1}}$. Then, from the definition of $I_i$, we know that $I_{q_1}, \cdots I_{q_{\lceil (n+1)q \rceil}} = 0$ (there could not be $\lceil (n+1)q \rceil$ values smaller than them), giving this property.

With these two properties, we could derive

$$\mathbb{E} \sum_{i=1}^{n+1} I_i = \sum_{i=1}^{n+1} \mathbb{E} I_i$$
$$= (n+1)\mathbb{E} I_{n+1} \quad \text{(property 1)}$$

and

$$\mathbb{E} \sum_{i=1}^{n+1} I_i \leq (n+1) - \lceil (n+1)q \rceil \quad \text{(property 2)}$$
$$\leq (n+1)(1-q).$$

Hence, $\mathbb{E} I_{n+1} \leq 1 - q$. From the definition of $\tau_{\text{MC}}$, we have

$$|\{i \mid S_i \leq \tau_{\text{MC}}\}| \geq \lceil n(1 + \frac{1}{n})q \rceil = \lceil (n+1)q \rceil.$$

Hence, $S_{n+1} > \tau_{\text{MC}}$ indicates $|\{i \mid S_i < S_{n+1}\}| \geq \lceil (n+1)q \rceil$, i.e. $I_{n+1} = 1$. Therefore,

$$\mathbb{P}(S_{n+1} \leq \tau_{\text{MC}}) = 1 - P(S_{n+1} > \tau_{\text{MC}})$$
$$\geq 1 - P(I_{n+1} = 1)$$
$$= 1 - \mathbb{E} I_{n+1} \geq q$$

gives us the conclusion. $\square$

**Remark.** *The i.i.d. condition could be replaced by a weaker condition of exchangeability. From this lemma, it could be seen that the i.i.d. property of scores is enough for the coverage guarantee of conformal prediction.*

**Proposition A.1.** *Under the conditions of Lemma 1, $\mathbb{P}[\hat{S}_{RS}(x_{n+1}, y_{n+1}) \leq \tau_{MC}] \geq 1 - \alpha + 2\beta$, where $\tau_{MC}$ is the $(1 - \alpha + 2\beta)(1 + 1/n)$-th empirical quantile of calibration scores. This is exactly Eq. (18)*

*Proof.* Let $q = 1 - \alpha + 2\beta$. Applying Lemma 1 and 2 gives the result. $\square$

### A.3 DISCUSSION ON THEOREM 1

In this section, we first introduce two concentration inequalities: Hoeffding's inequality and Empirical Bernstein's inequality (Maurer & Pontil, 2009). Then we explain how to utilize Hoeffding's inequality to derive bounds in the proof of Theorem 1, and how we could use Empirical Bernstein's inequality to improve the bound.

#### A.3.1 HOEFFDING'S INEQUALITY AND EMPIRICAL BERNSTEIN'S INEQUALITY

**Lemma A.3.** *(Hoeffding's Inequality) Let $X_1, \cdots, X_k$ be i.i.d. random variables bounded by the interval $[0, 1]$. Let $\overline{X} = \frac{1}{k} \sum_{j=1}^{k} X_j$, then for any $t \geq 0$*

$$\mathbb{P}(\overline{X} - \mathbb{E}\overline{X} \geq t) \leq e^{-2kt^2} \tag{A.1}$$

*and*

$$\mathbb{P}(\mathbb{E}\overline{X} - \overline{X} \geq t) \leq e^{-2kt^2}. \tag{A.2}$$

**Corollary A.4.** *From Hoeffding's Inequality, we argue that with probability at least $1 - \beta$,*

$$\mathbb{E}\overline{X} - \overline{X} \leq b_{Hoef}(\beta) = \sqrt{\frac{-log\beta}{2k}}. \tag{A.3}$$

$$\overline{X} - \mathbb{E}\overline{X} \leq b_{Hoef}(\beta) \tag{A.4}$$

*Proof.* Plugging $t = b_{\text{Hoef}}(\beta) = \sqrt{\frac{-log\beta}{2k}}$ into Eq. (A.2) gives

$$\mathbb{P}(\mathbb{E}\overline{X} - \overline{X} \geq b_{\text{Hoef}}) \leq \beta, \tag{A.5}$$

leading to Eq. (A.3) immediately. Similarly, we could get Eq. (A.4). $\square$

**Lemma A.5.** *(Empirical Bernstein Inequality) Under the condition of Lemma A.3, with probability at least $1 - \beta$,*

$$\overline{X} - \mathbb{E}\overline{X} \leq b_{Bern}(\beta, V) = \left[\sqrt{\frac{2V log\frac{2}{\beta}}{k}} + \frac{7log\frac{2}{\beta}}{3(k-1)}\right] \tag{A.6}$$

*where $V$ is the sample variance of $X_1, \cdots, X_k$, i.e.*

$$V = \frac{\sum_{j=1}^{k} X_j^2 - \frac{(\sum_{j=1}^{k} X_j)^2}{k}}{k-1}. \tag{A.7}$$

### A.3.2 APPLYING HOEFFDING'S INEQUALITY TO DERIVE EQ. (13) AND EQ. (14)

With Hoeffding's inequality, we can prove Eq. (13) and Eq. (14): Let $X_i = S(x_{n+1} + \delta_i, y_{n+1})$. Note that $\hat{S}_{\text{RS}}(x_{n+1}, y_{n+1}) = \frac{1}{N_{\text{MC}}} \sum_{i=1}^{N_{\text{MC}}} X_i = \overline{X}$ and $\mathbb{E}\overline{X} = S_{\text{RS}}(x_{n+1}, y_{n+1})$, we can apply Corollary A.4 and get

$$S_{\text{RS}}(x_{n+1}, y_{n+1}) - \hat{S}_{\text{RS}}(x_{n+1}, y_{n+1}) \leq b_{\text{Hoef}}(\beta), \tag{A.8}$$

with probability at least $1 - \beta$, which is exactly Eq. (13). Similarly, we could get

$$S_{\text{RS}}(\tilde{x}_{n+1}, y_{n+1}) \geq \hat{S}_{\text{RS}}(\tilde{x}_{n+1}, y_{n+1}) - b_{\text{Hoef}}(\beta) \tag{A.9}$$

with probability at least $1 - \beta$, which is exactly Eq. (14).

### A.3.3 USE EMPIRICAL BERNSTEIN'S INEQUALITY TO IMPROVE EQ. (14)

In practice, the predictor takes potentially perturbed input $\tilde{x}_{n+1}$ from the user, then generates a batch of Gaussian noise and computes corresponding $\hat{S}_{\text{RS}}$. Therefore, we could utilize the variation information from Monte Carlo samples to improve Eq. (14). Following the recommendation by (Zhai et al., 2020), we use Empirical Bernstein's inequality to provide a tighter bound. Below we show the details.

Let $X_i = S(\tilde{x}_{n+1} + \delta_i, y_{n+1})$. Note that $\hat{S}_{\text{RS}}(\tilde{x}_{n+1}, y_{n+1}) = \frac{1}{N_{\text{MC}}} \sum_{i=1}^{N_{\text{MC}}} X_i = \overline{X}$ and $\mathbb{E}\overline{X} = S_{\text{RS}}(\tilde{x}_{n+1}, y_{n+1})$, we can apply Lemma A.5 and get

$$\hat{S}_{\text{RS}}(\tilde{x}_{n+1}, y_{n+1}) - S_{\text{RS}}(\tilde{x}_{n+1}, y_{n+1}) \leq b_{\text{Bern}}(\beta, V), \tag{A.10}$$

with probability at least $1 - \beta$. With this inequality, we can derive Theorem A.6 and Corollary A.7 which are the counterparts of Theorem 1 and Corollary 2.

**Theorem A.6.** *Let $(x_{n+1}, y_{n+1})$ be the clean test sample and $\tilde{x}_{n+1}$ be perturbed input data that satisfies $\|\tilde{x}_{n+1} - x_{n+1}\|_2 \leq \epsilon$. Then, with probability $1 - 2\beta$:*

$$\hat{S}_{RS}(\tilde{x}_{n+1}, y_{n+1}) - b_{Bern}(\beta, V) \leq \Phi\left[\Phi^{-1}[\hat{S}_{RS}(x_{n+1}, y_{n+1}) + b_{Hoef}(\beta)] + \frac{\epsilon}{\sigma}\right],$$

*where $b_{Hoef}(\beta) = \sqrt{\frac{-ln\beta}{2N_{MC}}}$, $b_{Bern}(\beta, V) = \left[\sqrt{\frac{2V ln\frac{2}{\beta}}{N_{MC}}} + \frac{7ln\frac{2}{\beta}}{3(N_{MC}-1)}\right]$, $V$ is sample variance of $\hat{S}_{RS}$.*

**Corollary A.7.** *Let the prediction set*

$$C_\epsilon^+(\tilde{x}_{n+1}; \tau_{MC}) = \left\{k \in [K] \mid \hat{S}_{RS}(\tilde{x}_{n+1}, k) - b_{Bern}(\beta, V) \leq \Phi\left[\Phi^{-1}[\tau_{MC} + b_{Hoef}(\beta)] + \frac{\epsilon}{\sigma}\right]\right\}$$

*, $C_\epsilon^+(\tilde{x}_{n+1}; \tau_{MC})$ satisfies robust coverage guarantee in Eq. (2), i.e. $\mathbb{P}(y_{n+1} \in C_\epsilon^+(\tilde{x}_{n+1}; \tau_{MC})) \geq 1 - \alpha$.*

*Proof.* Replace Eq. (14) with Eq. (A.10), the remaining proofs are the same as the proof of Theorem 1 and Corollary 2. □

Compare Corollary 2 and Corollary A.7, we see that with Empirical Bernstein's inequality, we replace the $b_{\text{Hoef}}(\beta)$ on the left-hand side with $b_{\text{Bern}}(\beta, V)$, which could be computed at test time. It's natural to ask if we can also improve Eq. (13) with Empirical Bernstein's equality. Unfortunately, in practice, the predictor cannot access $x_{n+1}$ (the clean example corresponding to the input), hence we could not do Monte Carlo sampling and calculate variance to use Empirical Bernstein's equality. Therefore, we stick to Hoeffding's inequality for Eq. (13).

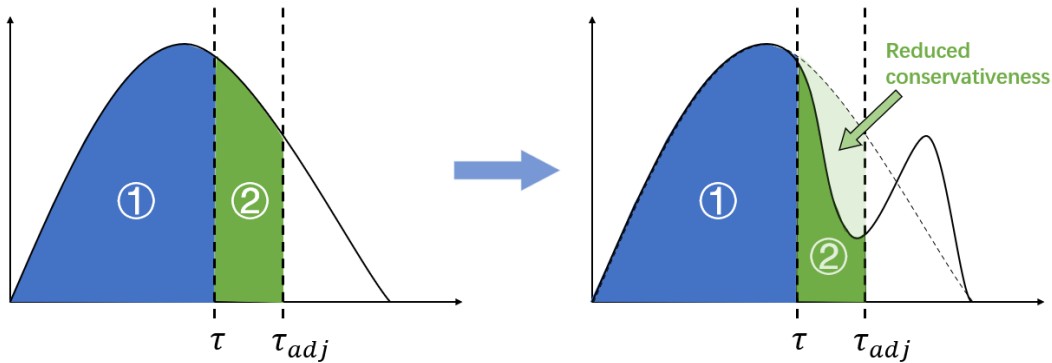

① **RSCP without threshold inflation**
② **Set inflation by threshold inflation ("conservativeness" in our manuscript)**

Figure A.1: **Density plot of (non)-conformity scores.** (Left) The prediction set of RSCP could be decomposed into two parts: (1) Base part generated by vanilla conformal prediction ($|C(x)|$, or RSCP before threshold inflation) and (2) Set inflation by threshold inflation (conservativeness). (Right) By reshaping the cdf, our method could reduce the second part, even without re-ranking the samples. This also gives an intuitive motivation for why we want to reduce the slope of cdf near the threshold (i.e. reduce the density near the threshold).

### A.4 MORE DISCUSSION ON CONSERVATIVENESS OF RSCP AND RSCP+

In Section 4, we motivate our PTT methods by discussing the conservativeness of the RSCP method. In this section, we provide a formal definition of conservativeness of RSCP and RSCP+, and discuss how our PTT and RCT methods could reduce this conservativeness.

**Definition A.1.** *For a conformal predictor that generates robust prediction set $C_\epsilon(x)$, we define its conservativeness as:*

$$cons. := \mathbb{E}_{x \sim P_x}(|C_\epsilon(x)| - |C(x)|), \tag{A.11}$$

*where $C(x)$ is the prediction set by vanilla conformal prediction generated with the same conformity score, and $P_x$ is the distribution of clean input $x$.*

**Remark.** *The predictor here could be RSCP or RSCP+. By this definition, conservativeness measures the increase in the average size of prediction sets, when we try to make the predictor robust to adversarial perturbations.*

With this definition, we could study the efficiency of robust conformal prediction methods by decomposing the average prediction set size into two parts: (1) The average size of vanilla conformal predictions $C(x)$, using the same conformity score and (2) the conservativeness. See Fig. A.1 where we give an intuitive illustration of these two parts. Fig. A.1 also provides an intuitive idea of why we should reduce the slope of score cdf (i.e. reducing the probability density as shown in the figure) near the threshold to reduce conservativeness.

**Empirical study.** We conduct an empirical study into these two parts on the CIFAR10 dataset, with HPS as the non-conformity score. The results are shown in Tab. A.1. In order to show a better comparison, we choose the RSCP benchmark because the baseline only gives trivial results on RSCP+. As shown in the table, our PTT method has a similar average size for the first part but reduced the conservativeness significantly. This verifies the intuition we get from Fig. A.1.

|                | Avg. Size of robust predictions $|C_\epsilon(x)|$ | Avg. of vanilla CP with randomized smoothing score $|C_{(}x)|$ | *cons.* |
|----------------|-----------|-----------|---------|
| Baseline       | 2.108     | 1.482     | 0.626   |
| PTT            | 1.779     | 1.468     | 0.311   |
| Size reduction | 0.329     | 0.014     | 0.315   |

Table A.1: Comparison of conservativeness: Our PTT method largely reduces the conservativeness of RSCP.

| Dataset      | CIFAR10 | CIFAR100 | ImageNet |
|--------------|---------|----------|----------|
| Average size | 10      | 100      | 1000     |

Table B.1: Baseline results with holdout set $D_{\text{holdout}}$ added to calibration set.

# B FURTHER DISCUSSION ON PTT METHOD IN SEC 4.1

## B.1 EXCHANGEBILITY OF RANK-TRANSFORMED SCORES

In the ranking transformation discussed in Sec. 4.1, we introduce the ranking transformation $\mathcal{Q}_{rank}$. A concern on this transformation will be, since both test scores and calibration scores rely on this holdout set, would the guarantee of conformal prediction be broken as they are no longer independent? Here, we show that the guarantee still holds as the ranking transformation keeps the exchangeability between scores. Denote $P_i = \mathcal{Q}_{rank}(S(x_i, y_i))$ as the calibration scores and $P_{n+1} = \mathcal{Q}_{rank}(S(x_{n+1}, y_{n+1}))$. We want to show that $P_1, P_2, \cdots, P_{n+1}$ are exchangeable. For any permutation $P_{i_1}, P_{i_2}, \cdots, P_{i_{n+1}}$, consider its pdf, we have

$$
p_{P_{i_1}, P_{i_2}, \cdots, P_{i_{n+1}}}(t_1, t_2, \cdots, t_{n+1})
$$
$$
= \int p_{P_{i_1}, P_{i_2}, \cdots, P_{i_{n+1}} | D_{\text{holdout}}}(t_1, t_2, \cdots, t_{n+1} \mid D) p_{D_{\text{holdout}}}(D) dD
$$
$$
= \int \prod_{j=1}^{n+1} p_{P_{i_j} | D_{\text{holdout}}}(t_j \mid D) p_{D_{\text{holdout}}}(D) dD
$$
$$
\text{(Given } D_{\text{holdout}} = D \text{, the scores become conditionally i.i.d.)}
$$
$$
= \int \prod_{j=1}^{n+1} p_{P_j | D_{\text{holdout}}}(t_j \mid D) p_{D_{\text{holdout}}}(D) dD
$$
$$
= p_{P_1, P_2, \cdots, P_{n+1}}(t_1, t_2, \cdots, t_{n+1}).
$$

(B.1)

Thus, after the ranking transformation, the scores are exchangeable, hence satisfy the condition of conformal prediction.

## B.2 CONCERNS ON ADDITIONAL DATA

As we discussed in Sec. 4.1, our PTT method requires an additional holdout set. In order to address the concern that our PTT benefits from using more data, we added the holdout set to the calibration set for the baseline method, so that the number of additional data samples will be the same for our PTT and baseline. With this modification, we found that the baseline still gave a trivial prediction set with all labels.

## B.3 RANKING TRANSFORMATION TURNS SCORE DISTRIBUTION INTO UNIFORM DISTRIBUTION

**Theorem B.8.** $(\mathcal{Q}_{rank} \circ S)(x, y)$ *is a discrete random variable, which takes values* $0, \frac{1}{|D_{holdout}|}, \frac{2}{|D_{holdout}|}, \cdots, 1$ *with equal probability.*

This result is well-known in statistics and we refer to Corollary 1 in Kuchibhotla (2020) that presented similar results, and Section 2.4 in Kuchibhotla (2020) for the proof.

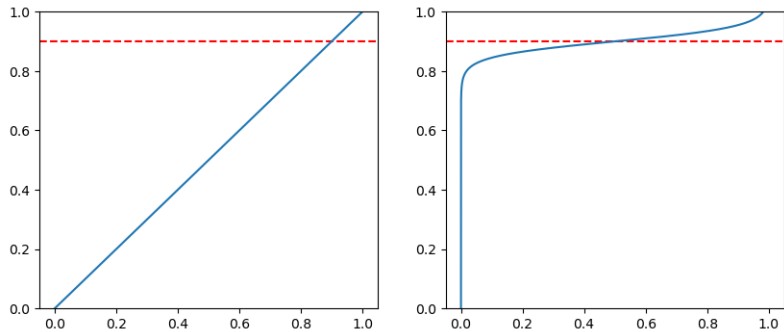

Figure B.1: **Comparison of score cdf before(left) and after(right) Sigmoid transformation.** The red dotted line denotes the desired coverage $1 - \alpha$. The left figure shows the cdf of uniformly distributed score $S$ after the ranking transformation. The right figure shows cdf of $\mathcal{Q}_{\text{sig}}(S)$. We could see that Sigmoid transformation greatly reduced the slope near threshold $\tau$ ($\tau = \Phi^{-1}(1 - \alpha)$, corresponding to the intersection of the red line and the blue curve).

## B.4  DESIGN CHOICE OF SIGMOID TRANSFORMATION

As we discussed in the main text, our goal is to design a monotonically increasing transformation that reduces the slope $\Phi'_S(\tau)$. After ranking transformation, the distribution of scores is turned into a uniform distribution, i.e. $\Phi_{\mathcal{Q}_{\text{rank}} \circ S}(t) = t$. Recall that $\mathcal{Q}_{\text{sig}}(s) = \phi\left[(s - b)/T\right]$. Thus, after applying our Sigmoid transformation, the cdf becomes

$$\Phi_{\mathcal{Q}_{\text{sig}} \circ \mathcal{Q}_{\text{rank}} \circ S}(t) = \mathcal{Q}_{\text{sig}}^{-1}(t) \tag{B.2}$$

and the derivative becomes

$$\Phi'_{\mathcal{Q}_{\text{sig}} \circ \mathcal{Q}_{\text{rank}} \circ S}(t) = (\mathcal{Q}_{\text{sig}}^{-1})'(t) = \left[\mathcal{Q}'_{\text{sig}}[\mathcal{Q}_{\text{sig}}^{-1}(t)]\right]^{-1} \tag{B.3}$$

The slope at the threshold is

$$
\begin{aligned}
\Phi'_{\mathcal{Q}_{\text{sig}} \circ \mathcal{Q}_{\text{rank}} \circ S}(\tau) &= \left[\mathcal{Q}'_{\text{sig}}[\mathcal{Q}_{\text{sig}}^{-1}(\tau)]\right]^{-1} \\
&= \left[\mathcal{Q}'_{\text{sig}}(1 - \alpha)\right]^{-1} \quad (\Phi_{\mathcal{Q}_{\text{sig}} \circ \mathcal{Q}_{\text{rank}} \circ S}(\tau) = \mathcal{Q}_{\text{sig}}^{-1}(\tau) = 1 - \alpha) \\
&= \frac{1}{T}\phi'\left[\frac{1 - \alpha - b}{T}\right] \quad (\text{Definition of } \mathcal{Q}_{\text{sig}})
\end{aligned}
\tag{B.4}
$$

By choosing $b = 1 - \alpha$, the right hand side equals $\frac{1}{4T}$. Hence, with $T$ large enough, we could achieve our goal of reducing $\Phi'_{\mathcal{Q}_{\text{sig}} \circ \mathcal{Q}_{\text{rank}} \circ S}(\tau)$ as discussed above.

## B.5  DISCUSSION ON MONOTONICITY CONDITION

In Sec. 4.1, we introduced two conditions for our desired transformation. One of these is the monotonicity condition which requires the transformation to be monotonically increasing. In this section, we discuss the reason we designed this condition and verify that the PTT we proposed satisfies this condition.

**Monotonicity ensures transformed scores keep the information.**  By applying the transformation, we hope the new score could become more robust to adversarial perturbations while maintaining the performance on clean examples. The monotonicity condition ensures this by the following theorem:

**Theorem B.9.** *For a monotonically non-decreasing transformation $\mathcal{Q}$, the new score $\mathcal{Q} \circ S$ is equivalent to $S$ in vanilla conformal prediction, i.e. they generate the same prediction set on all examples.*

*Proof.* Denote the threshold and prediction set generated by original score $S$ as $\tau_S$ and $C^S(x)$. Denote the threshold and prediction set generated by new score $\mathcal{Q} \circ S$ as $\tau_{\mathcal{Q} \circ S}$ and $C^{\mathcal{Q} \circ S}(x)$. From

the construction of the prediction set in vanilla conformal prediction (Eq. (3)), to show two prediction sets are identical, we only need to show

$$S(x, y) \leq \tau_S \Longleftrightarrow \mathcal{Q} \circ S(x, y) \leq \tau_{\mathcal{Q} \circ S} \tag{B.5}$$

Recall that the threshold is calculated as $(1 - \alpha)(1 + 1/|D_{\text{cal}}|)$ empirical quantile of calibration scores. Suppose for original score $S$, $\tau_S$ is the conformity score of $i$-th calibration example, i.e. $\tau_S = S(x_i, y_i)$. We argue that $\tau_{\mathcal{Q} \circ S}$ is also from $i$-th calibration example, i.e. $\tau_{\mathcal{Q} \circ S} = \mathcal{Q} \circ S(x_i, y_i)$, because a monotonically non-decreasing transformation does not change the order of calibration scores. Then Eq. (B.5) becomes

$$S(x, y) \leq S(x_i, y_i) \Longleftrightarrow \mathcal{Q} \circ S(x, y) \leq \mathcal{Q} \circ S(x_i, y_i) \tag{B.6}$$

This could be directly derived since $\mathcal{Q}$ is monotonically non-decreasing.

$\square$

**PTT satisfies monotonicity condition.** In order to verify this, we need to show both $\mathcal{Q}_{\text{rank}}$ and $\mathcal{Q}_{\text{sig}}$ are monotonically non-decreasing. For $\mathcal{Q}_{\text{rank}}$, rank function is non-decreasing. For $\mathcal{Q}_{\text{sig}}$, the Sigmoid function $\phi$ is non-decreasing, hence $\mathcal{Q}_{\text{sig}}$ is also non-decreasing for any $T > 0$.

### B.6 EMPIRICAL EVIDENCE FOR COVERAGE GAP REDUCTION BY PTT

In Sec. 4.1, we propose a transformation called PTT which satisfies slope reduction and monotonicity conditions. In Appendix B.7, we analyze a 1-D synthetic example and theoretically show that our PTT satisfying the two conditions could alleviate the conservativeness of RSCP. In this section, we study PTT empirically and show that our PTT satisfying these two conditions could reduce derivative $\Phi'_{\tilde{S}}(\tau)$, thus reduce coverage gap $\alpha_{\text{gap}}$. We apply PTT on the CIFAR10 dataset and compare the empirical CDF of $\tilde{S}$, as shown in Fig. B.2. To make comparison easier, we present the inverse of cdf. Note that

$$
\begin{aligned}
\Phi'_S(\tau) &= \Phi'_S[\Phi_S^{-1}(1 - \alpha)] && (\Phi_S(\tau) = 1 - \alpha) \\
&= [(\Phi_S^{-1})'(1 - \alpha)]^{-1}. && (f'(x) \cdot (f^{-1})'[f(x)] = 1)
\end{aligned}
\tag{B.7}
$$

Hence, we could compare $\Phi'_{\tilde{S}}(\tau)$ by comparing $(\Phi_S^{-1})'(1 - \alpha)$, the derivative of $\Phi_{\tilde{S}}^{-1}$ at target coverage $(1 - \alpha)$. Higher $\Phi_{\tilde{S}}^{-1}(1 - \alpha)$ means lower $\Phi'_{\tilde{S}}(\tau)$. As we could see in Fig. B.2, the transformed score $S_{\text{PTT}}$ has a higher derivative at $1 - \alpha = 0.9$, thus $\Phi'_{\tilde{S}}(\tau)$ is reduced.

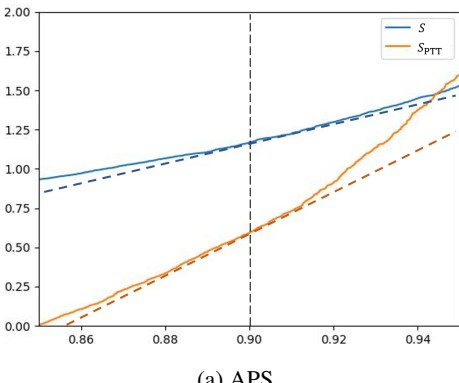

(a) APS

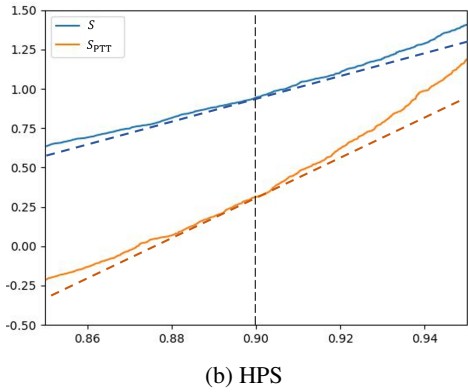

(b) HPS

Figure B.2: Empirical inverse cdf of $\tilde{S}$ on CIFAR10 test set, with base score HPS and APS. Comparing two base scores: the original score $S$ and our transformed score $S_{\text{PTT}}$, we see the derivative at $1 - \alpha = 0.9$ is larger for $S_{\text{PTT}}$, indicating its $\Phi'_{\tilde{S}}(\tau)$ is smaller.

## B.7 THEORETICAL ANALYSIS OF PTT ON A 1-D SYNTHETIC DATASET

In Sec. 4.1, we proposed PTT which applies a transformation on the base score to reduce conservativeness. In this section, we provide a theoretical analysis of our PTT method. Without any assumption on the data distribution, the analysis would be difficult. Hence, we construct a 1-D synthetic dataset and analyze our PTT method on it.

**Settings.** Consider a 1-D binary classification problem where $x \in [-0.5, 0.5]$ and $y \in \{-1, 1\}$. Suppose $\mathbb{P}(y = 1) = \mathbb{P}(y = -1) = 0.5$ and

$$x \sim \begin{cases} U(-0.5, 0), & y = -1; \\ U(0, 0.5), & y = 1. \end{cases}$$

Assume we use a linear model as the base model, where

$$\hat{\pi}(x, y = 1) = \begin{cases} x + 0.5, & x \in [-0.5, 0.5]; \\ 1, & x > 0.5; \\ 0, & x < -0.5. \end{cases}$$

and $\hat{\pi}(x, y = -1) = 1 - \hat{\pi}(x, y = 1)$. We study the HPS score here: $S_{\text{HPS}}(x, y) = 1 - \hat{\pi}(x, y)$

**Remark.** *Note that under this setting the score distribution $\Phi_{S_{HPS}}$ is uniform. The reason is the ranking transformation $\mathcal{Q}_{rank}$ involves a sampling process and makes the analysis difficult. Since the purpose of introducing $\mathcal{Q}_{rank}$ is to make the score uniformly distributed, we directly construct a uniformly distributed score, so that we can omit $\mathcal{Q}_{rank}$ in the following analysis.*

**Analysis.** Note that under this setting, the two classes $y = 1$ and $y = -1$ are actually symmetric. Therefore, in the analysis below, we focus on the class $y = -1$. Denote $h_S(x)$ as the smoothed score in Eq. (9) for class $y = -1$ with base score $S$:

$$\begin{aligned} h_S(x) &\triangleq \tilde{S}(x, y = -1; \sigma) \\ &= \Phi^{-1}[\mathbb{E}_{\delta \sim \mathcal{N}(0, \sigma^2)} S(x + \delta, y = -1)] \end{aligned} \tag{B.8}$$

In this case, we could derive a close form expression for $\alpha_{\text{gap}}$, $\Phi'_{\tilde{S}}(\tau)$ and average size.

**Theorem B.10.** *For any monotonically non-decreasing base score $S$, the coverage gap $\alpha_{gap}$ is*

$$\alpha_{gap} = \alpha + \Phi_{\tilde{S}}(\tau_{adj}) - 1.$$

*The average size is*

$$\begin{aligned} &\mathbb{E}_{(x,y) \sim P_{xy}} |C_\epsilon(x)| \\ &= \begin{cases} 2h_S^{-1}(\tau_{adj}) + 1, & \tau_{adj} \in (h_S(-0.5), h_S(0.5)); \\ 2, & \tau_{adj} \in [h_S(0.5), \infty). \end{cases} \end{aligned}$$

*The derivative is $\Phi'_{\tilde{S}}(\tau) = \frac{2}{h'_S(-\frac{\alpha}{2})}$.*

We defer the proof to Appendix B.7.1. With Theorem B.10, we could study how our transformation could help reduce average size. For $S = S_{\text{HPS}}$, we have $S_{\text{HPS}}(x, y = -1) = 1 - \hat{\pi}(x, y = -1) = \hat{\pi}(x, y = 1)$ and we could derive $h_{S_{\text{HPS}}}(x)$ and $h'_{S_{\text{HPS}}}(x)$ by Eq. (9). Next, we consider applying our transformation $\mathcal{Q}$ on HPS. Under simplifying assumptions, we get: $h_{S_{\text{PTT}}}(x) = \frac{x + 0.5 - b}{\sigma}$ and $h'_{S_{\text{PTT}}}(x) = \frac{1}{\sigma}$. For more details on our assumptions and the derivation process, please see Appendix B.7.1. Notice that both $S_{\text{HPS}}$ and our transformed score $S_{\text{PTT}}$ is monotonically increasing. Thus, we could apply Theorem B.10 by plugging in corresponding $S$ and derive the metrics we are interested in. We show results for different $\sigma$ in Tab. B.2.

**Conclusions** By this example, we could see

1. **The coverage gap $\alpha_{\text{gap}}$ reflects the conservativeness of RSCP.** As we could see in Tab. B.2, $\alpha_{\text{gap}}$ is a good indicator of conservativeness of predictor. By reducing $\alpha_{\text{gap}}$, we could make the average size smaller.

2. $\Phi'_{\tilde{S}}(\tau) \cdot M_\epsilon$ **is a good approximation for** $\alpha_{\textbf{gap}}$. In this case, $M_\epsilon = \frac{\epsilon}{\sigma}$. From Tab. B.2, we could see that $\alpha_{\text{gap}} \approx \Phi'_{\tilde{S}}(\tau) \cdot \frac{\epsilon}{\sigma}$, except for some cases where $\alpha_{\text{gap}}$ is upper-bounded by 10% (because $\alpha_{\text{gap}} \le \alpha$). This supports our linear approximation in Eq. (23).

3. **Our PTT could reduce the derivative** $\Phi'_{\tilde{S}}(\tau)$. Therefore, PTT is able to reduce the coverage gap $\alpha_{\text{gap}}$ and improve efficiency.

4. **By applying the Sigmoid transformation, the score converges to optimal score when** $T \to 0$. See Appendix B.7.2 for formal statement and proof.

| $\sigma^2$ | Base Score | $\Phi'_{\tilde{S}}(\tau) \cdot \frac{\epsilon}{\sigma}$ | $\alpha_{\text{gap}}$ | Avg. Size | Conservativeness |
|---|---|---|---|---|---|
| 0.01 | $S_{\text{HPS}}$ | 0.07916 | 7.98% | 0.98 | 0.08 |
| | $S_{\text{PTT}}$ | 0.02 | 2.00% | 0.92(-6.12%) | 0.02 |
| 0.001 | $S_{\text{HPS}}$ | 0.2503 | 10.00% | 1.15 | 0.25 |
| | $S_{\text{PTT}}$ | 0.02 | 2.00% | 0.92(-20.00%) | 0.02 |
| 0.0001 | $S_{\text{HPS}}$ | 0.7916 | 10.00% | 1.62 | 0.72 |
| | $S_{\text{PTT}}$ | 0.02 | 2.00% | 0.92(-43.21%) | 0.02 |

Table B.2: Results for the 1-D synthetic example, comparing HPS $S_{\text{HPS}}$ and our transformed score $S_{\text{PTT}}$. $\epsilon$ is set to 0.01 for all cases. The improvement relative to the baseline is shown in parentheses. From the results, it could be seen that our transformation consistently reduce $\alpha_{\text{gap}}$ and average set size for different $\sigma$.

### B.7.1 PROOFS

In this section, we discuss the details of the illustrative example. First, we present the derivation of Theorem B.10. Then we discuss how we apply Theorem B.10 to calculate metrics in Tab. B.2 for HPS $S_{\text{HPS}}$ and our transformed score $S_{\text{PTT}}$.

**Proof of Theorem B.10.** Recall that $h_S(x) \triangleq \tilde{S}(x, y = -1; \sigma)$.

**Lemma B.11.** *If base score $S$ is monotonically non-decreasing w.r.t. $x$, the $h_S(x)$ is monotonically non-decreasing w.r.t. $x$.*

*Proof.* Suppose we have $x_1 \le x_2$. From the monotonicity of $S$ and the fact that the smoothing operation $\mathbb{E}_{\delta \sim \mathcal{N}(0,\sigma^2)}$ as well as $\Phi^{-1}$ preserve the monotonicity, we have

$$S(x_1 + \delta, y = -1) \le S(x_2 + \delta, y = -1), \forall \delta \in \mathbb{R}$$
$$\Longrightarrow \mathbb{E}_{\delta \sim \mathcal{N}(0,\sigma^2)}\left[S(x_1 + \delta, y = -1)\right] \le \mathbb{E}_{\delta \sim \mathcal{N}(0,\sigma^2)}\left[S(x_2 + \delta, y = -1)\right]$$
$$\Longrightarrow h_S(x_1) \le h_S(x_2). \tag{B.9}$$

$\square$

**Lemma B.12.** *For any monotonically non-decreasing base score $S$, the cdf function of $\tilde{S}$ is:*

$$\Phi_{\tilde{S}}(t) = \begin{cases} 0, & t \in (-\infty, h_S(-0.5)]; \\ 2h_S^{-1}(t) + 1, & t \in (h_S(-0.5), h_S(0)); \\ 1, & t \in [h_S(0), \infty). \end{cases} \tag{B.10}$$

*Proof.*

$$\begin{aligned} \Phi_{\tilde{S}}(t) &= \mathbb{P}(\tilde{S} \le t) \\ &= \mathbb{P}(\tilde{S}(x, y = -1; \sigma) \le t \mid y = -1) && \text{(Symmetricity)} \\ &= \mathbb{P}(h_S(x) \le t \mid y = -1) \\ &= \mathbb{P}(x \le h_S^{-1}(t) \mid y = -1) && \text{(Monotonicity of } h(x)) \\ &= \begin{cases} 0, & t \in (-\infty, h_S(-0.5)]; \\ 2h_S^{-1}(t) + 1, & t \in (h_S(-0.5), h_S(0)); \\ 1, & t \in [h_S(0), \infty). \end{cases} && \text{(Distribution of } x) \end{aligned} \tag{B.11a}$$

$$\Box$$

With Lemmas B.11 and B.12, we can prove following results:

**Theorem B.10.** *For any monotonically non-decreasing base score $S$, the coverage gap $\alpha_{gap}$ is*

$$\alpha_{gap} = \alpha + \Phi_{\tilde{S}}(\tau_{adj}) - 1.$$

*The average size is*

$$\mathbb{E}_{(x,y)\sim P_{xy}}|C_\epsilon(x)|$$
$$= \begin{cases} 2h_S^{-1}(\tau_{adj}) + 1, & \tau_{adj} \in (h_S(-0.5), h_S(0.5)); \\ 2, & \tau_{adj} \in [h_S(0.5), \infty). \end{cases}$$

*The derivative is $\Phi'_{\tilde{S}}(\tau) = \frac{2}{h'_S(-\frac{\alpha}{2})}$.*

*Proof.* By Eq. (22) and Lemma B.12 and note that $0 < 1 - \alpha < 1$, we have

$$\Phi_{\tilde{S}}(\tau) = 1 - \alpha = 2h_S^{-1}(\tau) + 1, \tag{B.12}$$

which gives:

$$\tau = h_S(-\frac{\alpha}{2}). \tag{B.13}$$

The coverage gap $\alpha_{\text{gap}}$:

$$\begin{aligned}
\alpha_{\text{gap}} &= \alpha - \alpha_{\text{adj}} \\
&= \alpha - \mathbb{P}_{(x,y)\sim P_{xy}}(-1 \notin C_\epsilon(x) \mid y = -1) \qquad \text{(Symmetricity)} \\
&= \alpha - \mathbb{P}[\tilde{S}(x, y = -1) > \tau_{\text{adj}} \mid y = -1] \\
&= \alpha + \Phi_{\tilde{S}}(\tau_{\text{adj}}) - 1
\end{aligned} \tag{B.14}$$

and the average size $\mathbb{E}_{(x,y)\sim P_{xy}}|C_\epsilon(x)|$ can be calculated as:

$$\begin{aligned}
& \mathbb{E}_{(x,y)\sim P_{xy}}|C_\epsilon(x)| \\
=\ & \mathbb{P}(-1 \in C_\epsilon(x)) + \mathbb{P}(1 \in C_\epsilon(x)) \\
=\ & 2\mathbb{P}(h_S(x) \le \tau_{\text{adj}}) \\
=\ & 2\mathbb{P}(x \le h_S^{-1}(\tau_{\text{adj}})) \\
=\ & \begin{cases} 2h_S^{-1}(\tau_{\text{adj}}) + 1, & \tau_{\text{adj}} \in (h_S(-0.5), h_S(0.5)); \\ 2, & \tau_{\text{adj}} \in [h_S(0.5), \infty). \end{cases}
\end{aligned} \tag{B.15}$$

By Eqs. (B.12) and (B.13), the derivative can also be computed as

$$\begin{aligned}
\Phi'_{\tilde{S}}(\tau) &= 2(h_S^{-1})'(\tau) \\
&= \frac{2}{h'_S(-\frac{\alpha}{2})}.
\end{aligned} \tag{B.16a}$$

Eq. (B.16a) is derived by the derivative of the inverse function:

$$[f^{-1}](z) = \frac{1}{f'[f^{-1}(z)]}. \tag{B.17}$$

$$\Box$$

**Derivation of Tab. B.2.** With Theorem B.10, we could study related metrics for HPS $S_{\text{HPS}}$ and our transformed score $S_{\text{PTT}}$, only need to calculate $h_S(x)$ and $h'_S(x)$. By Eq. (9), we derive:

$$\begin{aligned}
& h_{S_{\text{HPS}}}(x) = \tilde{S}_{\text{HPS}}(x, y = -1; \sigma) \\
=\ & \Phi^{-1}\left[\mathbb{E}_{\delta\sim\mathcal{N}(0,\sigma^2)} S_{\text{HPS}}(x + \delta, y = -1)\right] \\
=\ & \Phi^{-1}\left[\frac{1}{\sqrt{2\pi\sigma^2}}\int \hat{\pi}(x + \delta, y = 1)e^{-\frac{\delta^2}{2\sigma^2}}\, d\delta\right] \\
=\ & \Phi^{-1}\left\{\frac{\sigma}{\sqrt{2\pi}}\left[e^{\frac{-(x+0.5)^2}{2\sigma^2}} - e^{\frac{-(x-0.5)^2}{2\sigma^2}}\right]\right. \\
& \left. + (x + 0.5)\left[\Phi(\frac{0.5 - x}{\sigma}) - \Phi(\frac{-0.5 - x}{\sigma})\right] + \Phi(\frac{x - 0.5}{\sigma})\right\}
\end{aligned} \tag{B.18}$$

$h'_{S_{\text{HPS}}}(x)$ could be computed by the chain rule:

$$
\begin{aligned}
&h'_{S_{\text{HPS}}}(x)\\
=\ & \frac{d}{dx}\Big\{\Phi^{-1}\big[\mathbb{E}_{\delta\sim\mathcal{N}(0,\sigma^2)}\,S_{\text{HPS}}(x+\delta, y=-1)\big]\Big\}\\
=\ & (\Phi^{-1})'\big[\mathbb{E}_{\delta\sim\mathcal{N}(0,\sigma^2)}\,S_{\text{HPS}}(x+\delta, y=-1)\big]\cdot\\
& \mathbb{E}_{\delta\sim\mathcal{N}(0,\sigma^2)}\left[\frac{d}{dx}S_{\text{HPS}}(x+\delta, y=-1)\right]\\
=\ & (\Phi^{-1})'\big[\mathbb{E}_{\delta\sim\mathcal{N}(0,\sigma^2)}\,S_{\text{HPS}}(x+\delta, y=-1)\big]\cdot\\
& \mathbb{E}_{\delta\sim\mathcal{N}(0,\sigma^2)}\left[\mathbb{1}_{\{-0.5\le x+\delta\le 0.5\}}\right]
\end{aligned}
\tag{B.19}
$$

Note that HPS satisfies the monotonicity condition, hence we could apply Theorem B.10 and Lemma B.12 by plugging in $h = h_{S_{\text{HPS}}}(x)$. Next, we consider how to apply our transformation $\mathcal{Q}$ on HPS. In order to simplify the analysis, we make two assumptions:

1. We skip the ranking transformation which involves sampling a holdout set, therefore $S_{\text{PTT}} = \mathcal{Q}_{\text{sig}} \circ S$. We argue that in this case the HPS is already uniformly distributed and the ranking step is not necessary.

2. In the sigmoid transformation, where $\mathcal{Q}_{\text{sig}} \circ S = \phi(\frac{S-b}{T})$, we let $T \to 0$. For the sigmoid function,

$$
\phi(x) = \frac{1}{1+e^{-x}} = \begin{cases}1, & x\to\infty\ (T\to 0, S > b);\\ 0, & x\to-\infty\ (T\to 0, S < b).\end{cases}
\tag{B.20}
$$

Therefore, by taking the limit $T \to 0$, the transformed score $S_{\text{PTT}}$ could be written as:

$$
S_{\text{PTT}}(x,y) = \begin{cases}0, & S_{\text{HPS}}(x,y) \le b,\\ 1, & S_{\text{HPS}}(x,y) > b.\end{cases}
\tag{B.21}
$$

With these assumptions, we could apply Eq. (9), which gives:

$$
\begin{aligned}
&h_{S_{\text{PTT}}}(x) = \tilde{S}_{\text{PTT}}(x, y=-1; \sigma)\\
=\ & \Phi^{-1}\big[\mathbb{E}_{\delta\sim\mathcal{N}(0,\sigma^2)}S_{\text{PTT}}(x+\delta, y=-1)\big]\\
=\ & \Phi^{-1}\big[\mathbb{E}_{\delta\sim\mathcal{N}(0,\sigma^2)}\mathbb{1}_{\{\hat{\pi}(x+\delta, y=1)\ge b\}}\big]\\
=\ & \Phi^{-1}\big[\mathbb{E}_{\delta\sim\mathcal{N}(0,\sigma^2)}\mathbb{1}_{\{\delta\ge b-x-0.5\}}\big]\\
=\ & \Phi^{-1}\big[\Phi(-\frac{b-x-0.5}{\sigma})\big]\\
=\ & \frac{x+0.5-b}{\sigma}
\end{aligned}
\tag{B.22}
$$

and

$$
h'_{S_{\text{PTT}}}(x) = \frac{1}{\sigma}.
\tag{B.23}
$$

Similar to $S_{\text{HPS}}$, $S_{\text{PTT}}$ is also monotonically increasing, thus we can apply Theorem B.10 and Lemma B.12 to get the results.

### B.7.2 OPTIMALITY OF OUR PTT

Below we show that in this case, our transformed score could achieve the smallest derivative $\Phi'_{\tilde{S}}(\tau)$ among all the base scores $S$ with the same threshold $\tau$.

**Theorem B.13.** *Among all monotonically increasing scores $S$ with the same threshold $\tau$ and $0 \le S(t, y=-1) \le 1$, $\Phi'_{\tilde{S}}(\tau)$ is minimized by $S(t; y=-1) = sgn(t + \frac{\alpha}{2})$, where $sgn(x) = \mathbf{1}\{x \ge 0\}$ is the sign function.*

*Proof.* From Theorem B.10, we could see that $\Phi'_{\tilde{S}}(\tau) = \frac{2}{h'_S(-\frac{\alpha}{2})}$. Hence, minimizing $\Phi'_{\tilde{S}}(\tau)$ is equivalent to maximizing $h'_S(-\frac{\alpha}{2})$. From definition of $h$, we have

$$
\frac{d}{dx}h_S(x) = \frac{d}{dx}\Phi^{-1}[\mathbb{E}_{\delta\sim\mathcal{N}(0,\sigma^2)}\ S(x+\delta, y=-1)]
$$
$$
= \frac{1}{\Phi'\left[\Phi^{-1}[\mathbb{E}_{\delta\sim\mathcal{N}(0,\sigma^2)}\ S(x+\delta, y=-1)]\right]}
$$
$$
\cdot \frac{d}{dx}\mathbb{E}_{\delta\sim\mathcal{N}(0,\sigma^2)}\ S(x+\delta, y=-1). \quad \text{(Chain rule, derivative of inverse function)}
$$

$$(B.24)$$

The second term

$$
\frac{d}{dx}\mathbb{E}_{\delta\sim\mathcal{N}(0,\sigma^2)}\ S(x+\delta, y=-1)
$$
$$
= \frac{d}{dx}\frac{1}{\sqrt{2\pi\sigma^2}}\int_{\mathbb{R}} e^{-\frac{\delta^2}{2\sigma^2}}S(x+\delta, y=-1)d\delta
$$
$$
= \frac{d}{dx}\frac{1}{\sqrt{2\pi\sigma^2}}\int_{\mathbb{R}} e^{-\frac{(x-t)^2}{2\sigma^2}}S(t, y=-1)dt \quad \text{(Let } t = x+\delta)
$$
$$
= \frac{1}{\sqrt{2\pi\sigma^2}}\int_{\mathbb{R}}\frac{d}{dx} e^{-\frac{(x-t)^2}{2\sigma^2}}S(t, y=-1)dt
$$
$$
= \frac{1}{\sqrt{2\pi\sigma^2}}\int_{\mathbb{R}}\frac{t-x}{\sigma^2} e^{-\frac{(x-t)^2}{2\sigma^2}}S(t, y=-1)dt
$$
$$
= \frac{1}{\sqrt{2\pi\sigma^2}}\left[\int_{-\infty}^{x}\frac{t-x}{\sigma^2} e^{-\frac{(x-t)^2}{2\sigma^2}}S(t, y=-1)dt + \int_{x}^{\infty}\frac{t-x}{\sigma^2} e^{-\frac{(x-t)^2}{2\sigma^2}}S(t, y=-1)dt\right]
$$
$$
\leq \frac{1}{\sqrt{2\pi\sigma^2}}\left[0 + \int_{x}^{\infty}\frac{t-x}{\sigma^2} e^{-\frac{(x-t)^2}{2\sigma^2}}dt\right] \quad (0 \leq S(t, y=-1) \leq 1)
$$

$$(B.25)$$

Note that the equality holds when $S(t, y=-1) = sgn(t-x)$. Plug in $x = -\frac{\alpha}{2}$, we get

$$
h'_S(-\frac{\alpha}{2}) = \frac{\frac{d}{dx}\mathbb{E}_{\delta\sim\mathcal{N}(0,\sigma^2)}\ S(-\frac{\alpha}{2}+\delta, y=-1)}{\Phi'\left[\Phi^{-1}[\mathbb{E}_{\delta\sim\mathcal{N}(0,\sigma^2)}\ S(-\frac{\alpha}{2}+\delta, y=-1)]\right]}.
$$
$$
= \frac{\frac{d}{dx}\mathbb{E}_{\delta\sim\mathcal{N}(0,\sigma^2)}\ S(-\frac{\alpha}{2}+\delta, y=-1)}{\Phi'\left[h_S(-\frac{\alpha}{2})\right]}.
$$

$$(B.26)$$

Recall that $\tau = h_S(-\frac{\alpha}{2})$. (Eq. (B.13)), the denominator is just $\Phi'(\tau)$, which is a constant given $\tau$. The numerator is maximized by $S(t, y=-1) = sgn(t+\frac{\alpha}{2})$, giving the conclusion. $\qquad\square$

**Remark.** *Note that the optimal base score is attained by applying our transformation on the HPS score, with $b = \frac{1-\alpha}{2}$ and $T \to 0$, as in Eq. (B.20). This supports the asymptotic optimality of our transformation.*

### B.8 CASE STUDY: POTENTIAL FAILURE MODE OF PTT

A natural question would be: when will the PTT fail to boost the efficiency? Here, we provide a special case where PTT will fail to boost efficiency. Suppose the data distribution is the same as we defined in the settings part of Appendix B.7. For the base score, let

$$
S_b(x, y=-1) = \begin{cases} 0.5 + o(1), & x \leq -0.5; \\ x + 0.5, & x \in (-0.5, 0.5); \\ 0.5 - o(1), & x \geq 0.5, \end{cases}
$$

$$(B.27)$$

and $S_b(x, y=1) = 1 - S_b(x, y=-1)$. That is to say, the conformity score is slightly in favor of the wrong class when $x \notin (-0.5, 0.5)$. Here, we introduce an ignorable small deviation ($o(1)$) so that

the score will be above(below) the PTT threshold $b$ for $x \leq -0.5$ and $x \geq 0.5$. Let $1 - \alpha = b = 0.5$, by Eq. (B.21), the PTT score will be

$$S_{\text{PTT}} = \begin{cases} 1, & x \in (-\infty, -0.5] \cup [0, 0.5); \\ 0, & x \in (-0.5, 0) \cup [0.5, \infty). \end{cases} \tag{B.28}$$

Similar to Appendix B.7.1, we could derive the smoothed scores:

$$
\begin{aligned}
h_{S_b}(x) &= \tilde{S}_b(x, y = -1) \\
&= \Phi^{-1} \Bigg\{ \frac{\sigma}{\sqrt{2\pi}} \left[ e^{\frac{-(x+0.5)^2}{2\sigma^2}} - e^{\frac{-(x-0.5)^2}{2\sigma^2}} \right] \\
&\quad + (x + 0.5) \left[ \Phi(\frac{0.5 - x}{\sigma}) - \Phi(\frac{-0.5 - x}{\sigma}) \right] + \frac{1}{2}\Phi(\frac{x - 0.5}{\sigma}) + \frac{1}{2}\Phi(\frac{-x - 0.5}{\sigma}) \Bigg\},
\end{aligned}
\tag{B.29}
$$

and

$$
\begin{aligned}
h_{S_{\text{PTT}}}(x) &= \tilde{S}_{\text{PTT}}(x, y = -1) \\
&= \Phi^{-1} \big[ \mathbb{E}_{\delta \sim \mathcal{N}(0,\sigma^2)} \mathbb{1}_{\{x+\delta \leq -0.5\}} + \mathbb{1}_{\{0 \leq x+\delta \leq 0.5\}} \big] \\
&= \Phi^{-1} \big[ \mathbb{E}_{\delta \sim \mathcal{N}(0,\sigma^2)} \mathbb{1}_{\{\delta \leq -x-0.5\}} + \mathbb{1}_{\{-x \leq \delta \leq -x+0.5\}} \big] \\
&= \Phi^{-1} \left[ \Phi(\frac{-x - 0.5}{\sigma}) + \Phi(\frac{-x + 0.5}{\sigma}) - \Phi(\frac{-x}{\sigma}) \right]
\end{aligned}
$$

Since the scores are no longer monotonically increasing, we cannot directly apply Theorem B.10 to get a closed-form solution. However, we could numerically calculate the CDF $\Phi_{\tilde{S}}$, threshold $\tau$ and the average size $C_\epsilon(x)$ as we know the smoothed score function. We choose the inflation level $\frac{\epsilon}{\sigma} = 0.01$ across experiments, and report the results in Tab. B.3. The results show that for $\sigma = 0.3$, the PTT method is worse than the original score. This indicates that theoretically, for a specific base score, the PTT method could amplify the error and produce a larger set.

However, this kind of failure mode is not observed in our experiments in Sec. 5. We argue that despite being theoretically possible, the failure case shown in this section requires (1) a carefully designed base score and (2) a specific choice of $1 - \alpha$ and $\sigma$. Hence, the chance that it happens in practice is rare.

| $\sigma$ | Base Score | $\tau$ | Avg. Size | Conservativeness |
|---|---|---|---|---|
| 0.1 | $S_b$ | -0.620 | 0.512 | 0.012 |
| | $S_{\text{PTT}}$ | -1.245 | **0.504** | 0.004 |
| 0.2 | $S_b$ | -0.464 | 0.529 | 0.029 |
| | $S_{\text{PTT}}$ | -0.536 | **0.511** | 0.011 |
| 0.3 | $S_b$ | -0.302 | **0.525** | 0.025 |
| | $S_{\text{PTT}}$ | -0.169 | 0.529 | 0.029 |

Table B.3: Results for the failure case study. Note that the average size increased after PTT when $\sigma = 0.3$.

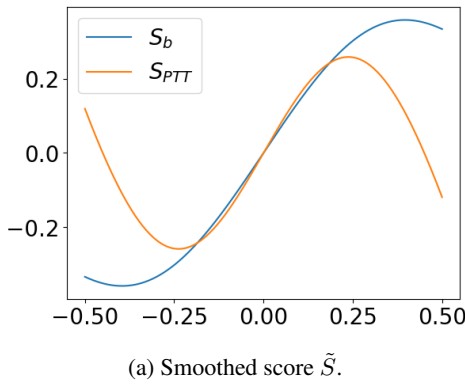 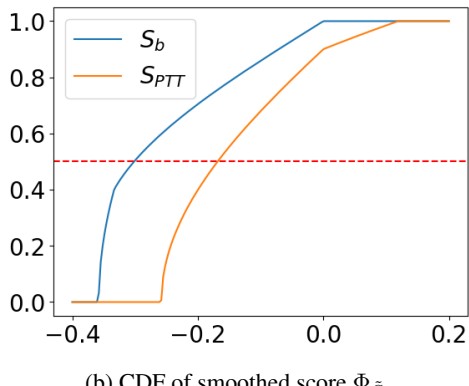

(a) Smoothed score $\tilde{S}$.           (b) CDF of smoothed score $\Phi_{\tilde{S}}$.

Figure B.3: Comparison between original score $S_b$ and PTT score $S_{\text{PTT}}$ when $\sigma = 0.3$. (Left) Smoothed scores. (Right) CDF of smoothed scores. The red dotted line denotes the coverage level $1 - \alpha = 0.5$. The intersection between the dotted line and the CDF curve is the threshold $\tau$. From the figure, we could intuitively see that the CDF of the PTT score has a larger derivative at the threshold.

## C    FURTHER DISCUSSION ON ROBUST CONFORMAL TRAINING IN SEC 4.2

### C.1    DETAILS OF CONFORMAL TRAINING (STUTZ ET AL., 2021)

Conformal training (Stutz et al., 2021) simulates calibration and prediction process on mini-batches during training. In training, each batch $B$ is split into two subsets: calibration set $B_{\text{cal}}$ and prediction set $B_{\text{pred}}$. The threshold $\tau$ is calculated based on $B_{\text{cal}}$, and the prediction sets $C(x; \tau)$ are generated for samples in $B_{\text{pred}}$. Under this setting, $S$ is a function of trainable parameters $\theta$. To emphasize this, we denote the base score as $S_\theta(x, y)$ in this section. In order to perform back-propagation to update $\theta$, Stutz et al. (2021) proposed to calculate soft threshold $\tau^{\text{soft}}$ and soft prediction set $c(x, y; \tau^{\text{soft}})$ as differentiable approxiamation of threshold $\tau$ and prediction set $C(x)$. We discuss the details next.

**Soft threshold $\tau^{\text{soft}}$**    The calculation of threshold $\tau$ involves calculating a specific quantile of a given set, which is not differentiable. To address this problem, smooth sorting methods (Blondel et al., 2020; Cuturi et al., 2019) are utilized to get a smoothed version of quantile:

$$\tau^{\text{soft}} = Q_{1-\alpha}^{\text{soft}}(\{S_\theta(x, y)\}_{(x,y) \in B_{\text{cal}}}), \tag{C.1}$$

where $Q_q^{\text{soft}}(H)$ denotes the $q(1 + \frac{1}{|H|})$-quantile of set $H$ derived by smooth sorting.

**Soft prediction set $c(x, y; \tau^{\text{soft}})$**    In the prediction stage, Stutz et al. (2021) introduced a class-wise prediction function $c(x, y) \in [0, 1]$ which represents the probability that label $y$ is included in the prediction set of $x$. The prediction set could now be represented as the collection of class-wise predictions: $C(x) = \{y \in [K] \mid c(x, y)\}$. Using this notation, the vanilla conformal prediction (Angelopoulos et al., 2020) could be expressed as:

$$c_{\text{hard}}(x, y; \tau) = \begin{cases} 1, & S_\theta(x, y) \leq \tau; \\ 0, & S_\theta(x, y) > \tau. \end{cases}$$

which requires to perform hard-thresholding w.r.t. threshold $\tau$. Stutz et al. (2021) proposed to replace $c_{hard}$ by a smooth version:

$$c(x, y; \tau^{\text{soft}}) = \phi \left[ \frac{\tau^{\text{soft}} - S_\theta(x, y)}{T_{\text{train}}} \right], \tag{C.2}$$

where $\phi(z) = 1/(1 + e^{-z})$ is the sigmoid function and temperature $T_{\text{train}}$ is a hyper-parameter.

## C.2 DETAILS OF ROBUST CONFORMAL TRAINING (RCT)

In this section, we discuss some design details of Robust Conformal Training (RCT). We first introduce some problems suggested by Stutz et al. (2021) and show how we avoid those problems.

Different from the conformal prediction method we introduce in Sec. 2.1, Stutz et al. (2021) used a slightly different convention, where $C(X; \tau') = \{k \in [K] \mid S'(x, y) \geq \tau'\}$ and threshold $\tau'$ is calculated as the $\alpha(1 + 1/|D_{\text{cal}}|)$-quantile of calibration scores. For example, $S'(x, y) = \hat{\pi}_y(x)$ for THR (Sadinle et al., 2019) and $S'(x, y) = \log \hat{\pi}_y(x)$ for THRLP. We argue that these two conventions are actually equivalent. Let $S(x, y) = -S'(x, y)$ and $\tau = -\tau'$, we see now $\tau$ is the $(1 - \alpha)(1 + 1/|D_{\text{cal}}|)$-quantile of calibration scores and $C(X; \tau) = \{k \in [K] \mid S(X, k) \leq \tau\}$, which is consistent with the notations we introduce. For the consistency of notations, we make some modifications to the notations in Stutz et al. (2021) by swapping their signs in the rest of this section.

Regarding the choice of (non-)conformity score for training, Stutz et al. (2021) mentioned a gradient diminishing problem when using THR: $S_{\text{THR}} = -\hat{\pi}_y(x)$. Thus, they proposed to add a logarithm function to mitigate this problem, i.e. using THRLP: $S_{\text{THRLP}} = -\log \hat{\pi}_y(x)$. However, this choice doesn't work when we try to incorporate RSCP into the scheme: RSCP requires the base score to take value in $[0, 1]$, which is not true for $S_{\text{THR}}$ and $S_{\text{THRLP}}$. To solve this problem, we choose HPS as the base score: since the difference between HPS and THR is only a constant, these two scores are actually equivalent in the context of conformal training (Stutz et al., 2021). HPS takes value in the interval $[0, 1]$, which satisfies the requirement of RSCP. A potential problem with this choice is: as suggested by Stutz et al. (2021), using HPS as the base score would cause gradient diminishing. Fortunately, we observe that the Gaussian inverse cdf step $\Phi^{-1}$ plays a similar role to the logarithm function in Stutz et al. (2021), which alleviates the diminishing gradient problem and leads to successful training.

# D   EXPERIMENTAL DETAILS AND EXTENSIVE STUDIES IN SEC 5

In this section, we discuss more details of our experiments in Sec. 5, and provide extensive empirical studies on the `RSCP+` and PTT / RCT methods we proposed.

## D.1   EXPERIMENTAL DETAILS

**Robust conformal training.**   For robust conformal training, as Stutz et al. (2021) suggested, we freeze the backbone of the base model and train the last linear layer after randomly reinitializing it. Regarding the base score for training, see Appendix C.2 which discusses our choice. For CIFAR10 and CIFAR100, standard data augmentation is applied which includes random flips and crops. In training, we use SGD with momentum 0.9 and Nesterov gradient. Weight decay is set to 0.0005. We finetune the model for $N_{\mathrm{epoch}} = 150$ epochs and scale the learning rate down by 0.1 at epoch 60, 90 and 120. In Eq. (25), we choose $N_{\mathrm{train}} = 8$. For other hyper-parameters, see Tab. D.1.

| Dataset | Batch size | Learning rate | $T_{\mathrm{train}}$ in Sec. 4.2 | Size weight | $\kappa$ in Eq. (26) |
|---------|-----------|---------------|------------|-------------|------------|
| CIFAR10 | 500 | 0.05 | 0.1 | 1 | 1 |
| CIFAR100 | 100 | 0.005 | 1 | 0.01 | 1 |

Table D.1: Hyperparameters setting for robust conformal training on CIFAR10 and CIFAR100.

**Dataset split**   For RCT training, the training set of CIAFR10 and CIFAR100 is utilized. For evaluation, we split the validation set into three subsets: $D_{\mathrm{holdout}}$ for ranking transformation in Sec. 4.1, $D_{\mathrm{cal}}$ for calibration, and $D_{\mathrm{test}}$ for evaluation of results. For the size of each subset, please refer to Tab. D.2. We want to point out one thing that in all experiments we employ the random calibration-test split suggested by Gendler et al. (2021). That means we fix training set $D_{\mathrm{train}}$ and holdout set $D_{\mathrm{holdout}}$, while randomly split the remaining data into $D_{\mathrm{cal}}$ and $D_{\mathrm{test}}$ for $N_{\mathrm{split}} = 50$ times. For each random split, calibration is performed on $D_{\mathrm{cal}}$, and prediction results are evaluated on $D_{\mathrm{test}}$. For a fair comparison, we add the holdout set to the calibration set for the baseline method (i.e. calibration is performed on $D_{\mathrm{holdout}} \cup D_{\mathrm{cal}}$), as they do not need this holdout set. All the metrics we present are the average of $N_{\mathrm{split}} = 50$ experiments.

| Dataset | training | holdout | calibration | test |
|---------|----------|---------|-------------|------|
| CIFAR10 | 50000 | 500 | 4750 | 4750 |
| CIFAR100 | 50000 | 500 | 4750 | 4750 |
| ImageNet | - | 500 | 24750 | 24750 |

Table D.2: Spilt of each dataset. Note that on ImageNet we only experiment PTT method which is training-free, therefore, the training set for ImageNet is omitted.

## D.2   EMPIRICAL RESULTS ON RSCP BENCHMARK

In this section, we present experiment results under the original RSCP benchmark (Gendler et al., 2021). As we point out in the main text, their approach is flawed in robustness guarantee, but it still serves as a simple empirical benchmark. For empirical robustness with RSCP, we follow the evaluation of Gendler et al. (2021) which uses SmoothAdv (Salman et al., 2019) to generate adversarial examples and examines the coverage and average size on these examples.

**Results.**   The results are shown in Tabs. D.3 and D.4. First, we could see that all methods achieve coverage above $1 - \alpha = 90\%$ under SmoothAdv attack, suggesting that our methods preserve the empirical robustness of Gendler et al. (2021). Compared with the baseline, our methods are closer to vanilla CP and have a smaller coverage gap $\alpha_{\mathrm{gap}}$. Next, focusing on average prediction set size, which is the key metric evaluating the efficiency of conformal prediction, our methods provide reduction up to 31.12%, 25.80%, and 48.80% on CIFAR10, CIFAR100, and ImageNet respectively. Note that our PTT method provides the first result on the ImageNet dataset to improve the size of robust conformal prediction *without* training. The conformal training methods in prior work (Einbinder et al., 2022b;

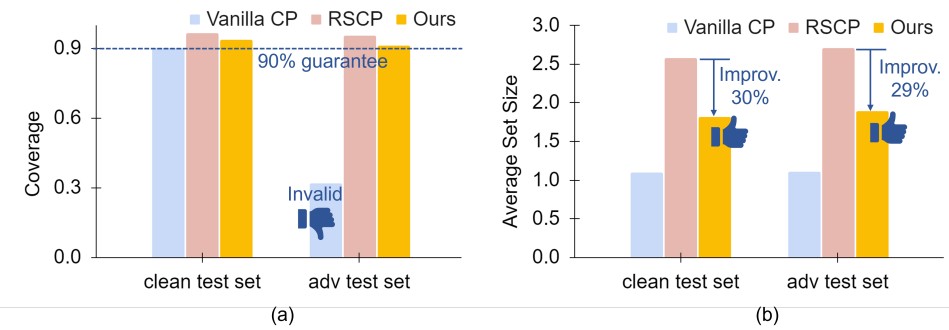

Figure D.1: Comparison of vanilla conformal prediction (Vanilla CP) (Romano et al., 2020), vanilla RSCP (Gendler et al., 2021) and our proposed method PTT + RCT with RSCP on (a) coverage guarantees and (b) average prediction set size on CIFAR10. Our methods achieve better (smaller) average set size than RSCP while satisfying coverage guarantees on the adversarial test set. Note that Vanilla CP does not achieve the desired 90% coverage guarantees on the adversarial test set. The non-conformity score applied here is APS (Romano et al., 2020).

| | CIFAR10 | | | | CIFAR100 | | | |
| | APS | | HPS | | APS | | HPS | |
| Method | Coverage | Average size ↓ | Coverage | Average size ↓ | Coverage | Average size ↓ | Coverage | Average size ↓ |
|---|---|---|---|---|---|---|---|---|
| Baseline (Gendler et al., 2021) | 95.68% | 2.751 | 93.54% | 2.108 | 93.53% | 16.19 | 93.43% | 14.30 |
| PTT (Ours) | 92.06% | 2.202 (-19.96%) | 90.90% | **1.779 (-15.61%)** | 91.26% | 12.78 (-21.06%) | 90.87% | 10.78 (-24.62%) |
| PTT + RCT (Ours) | 91.15% | **1.895 (-31.12%)** | 91.19% | 1.864 (-11.57%) | 91.07% | **12.32 (-23.88%)** | 90.83% | **10.61 (-25.80%)** |

Table D.3: **Coverage and average size on CIFAR 10 and CIFAR 100 with RSCP.** Results are presented on adversarial examples generated by SmoothAdv(Salman et al., 2019) with magnitude $\epsilon = 0.125$. The target coverage is $1 - \alpha = 0.9$ and the smoothing strength $\sigma = 0.25$. The improvement relative to the baseline is shown in parentheses and the best results are bold. Our method consistently gives prediction sets with smaller sizes, and it can be seen that the robustness of RSCP holds empirically.

| | APS | | HPS | |
| Method | Coverage | Average size ↓ | Coverage | Average size ↓ |
|---|---|---|---|---|
| Baseline(Gendler et al., 2021) | 95.36% | 51.72 | 94.12% | 16.64 |
| PTT (Ours) | 91.17% | **30.20 (-41.61%)** | 90.53% | **8.52 (-48.80%)** |

Table D.4: **Coverage and average set size results for ImageNet with RSCP.** Results are presented on adversarial example with magnitude $\epsilon = 0.25$, target coverage $1 - \alpha = 0.9$ and $\sigma = 0.5$. The improvement relative to the baseline is shown in parentheses.

Stutz et al., 2021) require training, which might be costly when the base model is large and does not scale to larger datasets like ImageNet.

| | RSCP | | | RSCP+ | | |
| $\epsilon$ | 0.125 | 0.25 | 0.5 | 0.125 | 0.25 | 0.5 |
|---|---|---|---|---|---|---|
| Baseline | 2.108 | 3.303 | 5.759 | 10 | 10 | 10 |
| PTT | **1.779** | **2.578** | **4.222** | **2.152** | **3.269** | **5.559** |
| PTT+RCT | 1.864 | 2.642 | 4.245 | **2.152** | 3.413 | 5.603 |

Table D.5: **Average size for different $\epsilon$ on CIFAR10 dataset, with RSCP and RSCP+.** Our methods consistently improve over the baseline.

|  | RSCP | | | | RSCP+ | | | |
|---|---|---|---|---|---|---|---|---|
| $\sigma$ | 0.125 | 0.25 | 0.5 | 1 | 0.125 | 0.25 | 0.5 | 1 |
| Baseline | 1.999 | 2.108 | 2.573 | 3.78 | 10 | 10 | 10 | 10 |
| PTT | **1.622** | **1.779** | **2.244** | **3.434** | 3.341 | **2.152** | **2.916** | **4.663** |
| PTT+RCT | 1.705 | 1.864 | 2.296 | 3.473 | **3.282** | **2.152** | 3.059 | 4.735 |

Table D.6: **Average size for different $\sigma$ on CIFAR10 dataset, with RSCP and RSCP+.** Our methods consistently improve over the baseline.

## D.3 EXPERIMENTS WITH DIFFERENT $\epsilon$ AND $\sigma$.

In this section, we extend our experiment on the CIFAR10 dataset to different $\epsilon$ and $\sigma$. We conduct experiments under two settings: (1) We fixed $\frac{\epsilon}{\sigma} = \frac{1}{2}$, and used $\epsilon = 0.125, 0.25, 0.5$. The results are in Tab. D.5 (2) We fixed $\epsilon = 0.125$ and tried $\sigma = 0.125, 0.25, 0.5, 1$. The results are in Tab. D.6. Both experiments are carried out on the CIFAR10 dataset with HPS score. For the RSCP+ benchmark, we apply the Empirical Bernstein's improvement introduced in Appendix A.3.3. Other hyperparameters are the same as the experiments we present in Sec. 5. From Tabs. D.5 and D.6, we could see that our methods consistently give a smaller average size, compared with the baseline (Gendler et al., 2021). This supports our claim that our methods could improve the efficiency, for both RSCP and RSCP+.

## D.4 ABALATION STUDY

In this section, we study the necessity of robust conformal training introduced in Sec. 4.2 by comparing it with Conformal Training (ConfTr)(Stutz et al., 2021). The experiments are carried out using the original RSCP benchmark, so we introduce RSCP (Gendler et al., 2021) as a baseline. As illustrated in Tab. D.7, ConfTr generates up to 66.08% larger prediction set than no conformal training, which is undesired, while with our RCT, we can reduce the size up to 17.52%, which supports the necessity of our RCT method. The reason may be that conformal training is not specifically crafted for randomized smoothing.

|  | APS | | HPS | |
|---|---|---|---|---|
| Training Method | Coverage | Average size ↓ | Coverage | Average size ↓ |
| Baseline(Cohen et al., 2019) | 95.68% | 2.751 | 93.54% | 2.108 |
| ConfTr(Stutz et al., 2021) | 90.21% | 3.509 (+27.55%) | 90.39% | 3.501 (+66.08%) |
| RCT (Ours) | 93.15% | **2.269 (-17.52%)** | 91.25% | **1.929 (-8.49%)** |

Table D.7: **Ablation study:** We compare our robust conformal prediction (RCT) with conformal training (ConfTr) (Stutz et al., 2021) on CIFAR10 dataset. It could be seen that applying ConfTr to train a base classifier for RSCP results in performance even worse than the baseline which uses the original weights by Cohen et al. (2019), supporting the necessity of our method.

## D.5 STANDARD DEVIATION OF EXPERIMENT RESULTS

In this section, we present the standard deviation of our experiment results.

| Base score | HPS | | APS | |
|---|---|---|---|---|
| Dataset | CIFAR10 | CIFAR100 | CIFAR10 | CIFAR100 |
| Baseline (Gendler et al., 2021) | 10 | 100 | 10 | 100 |
| PTT (**Ours**) | **2.294 $\pm$ 0.051** | $26.06 \pm 15.48$ | **2.685 $\pm$ 0.037** | $21.96 \pm 1.055$ |
| PTT+RCT (**Ours**) | **2.294 $\pm$ 0.051** | **18.30 $\pm$ 1.08** | 2.824 $\pm$ 0.047 | **20.01 $\pm$ 0.94** |

Table D.8: **Average size of RSCP+ on CIFAR10 and CIFAR100. with standard deviation.** For CIFAR10 and CIFAR100, $\epsilon = 0.125$ and $\sigma = 0.25$. For the large standard deviation with PTT on CIFAR10 using HPS as a base score, the reason is the predictor gives a trivial prediction during 50 tests. We argue that the probability is small and could be alleviated by applying RCT or increasing the number of Monte Carlo samples.

| Base score | HPS | APS |
|---|---|---|
| Baseline (Gendler et al., 2021) | 1000 | 1000 |
| PTT **(Ours)** | $1000 \pm 0$ | $94.66 \pm 130.97$ |
| PTT + Bernstein **(Ours)** | $\mathbf{59.12} \pm \mathbf{135.87}$ | $\mathbf{70.87} \pm \mathbf{3.69}$ |

Table D.9: **Average size of `RSCP+` on ImageNet. with standard deviation.** For ImageNet, $\epsilon = 0.25$ and $\sigma = 0.5$. Some entries in the table have large standard deviations. The reason is the predictor gives a trivial prediction during 50 tests. We argue that the probability is small and could be alleviated by increasing the number of Monte Carlo samples.

| | APS | | HPS | |
|---|---|---|---|---|
| Method | Coverage | Average size ↓ | Coverage | Average size ↓ |
| Baseline (Gendler et al., 2021) | $95.68\% \pm 0.30\%$ | $2.751 \pm 0.031$ | $93.54\% \pm 0.48\%$ | $2.108 \pm 0.036$ |
| PTT (Ours) | $92.06\% \pm 0.40\%$ | $2.202 \pm 0.029$ | $90.90\% \pm 0.61\%$ | $\mathbf{1.779} \pm 0.029$ |
| PTT + RCT (Ours) | $91.15\% \pm 0.48\%$ | $\mathbf{1.895} \pm 0.033$ | $91.19\% \pm 0.51\%$ | $1.864 \pm 0.027$ |

Table D.10: Coverage and average size on CIFAR 10 with RSCP with standard deviation.

| | APS | | HPS | |
|---|---|---|---|---|
| Method | Coverage | Average size ↓ | Coverage | Average size ↓ |
| Baseline (Gendler et al., 2021) | $93.53\% \pm 0.44\%$ | $16.19 \pm 0.47$ | $93.43\% \pm 0.47\%$ | $14.30 \pm 0.43$ |
| PTT (Ours) | $91.26\% \pm 0.55\%$ | $12.78 \pm 0.34$ | $90.87\% \pm 0.61\%$ | $10.78 \pm 0.36$ |
| PTT + RCT (Ours) | $91.07\% \pm 0.59\%$ | $\mathbf{12.32} \pm 0.38$ | $90.83\% \pm 0.58\%$ | $\mathbf{10.61} \pm 0.32$ |

Table D.11: Coverage and average size on CIFAR 100 with RSCP with standard deviation.

| | APS | | HPS | |
|---|---|---|---|---|
| Method | Coverage | Average size ↓ | Coverage | Average size ↓ |
| Baseline (Gendler et al., 2021) | $95.36\% \pm 0.14\%$ | $51.72 \pm 0.819$ | $94.12\% \pm 0.21\%$ | $16.64 \pm 0.387$ |
| PTT (Ours) | $91.17\% \pm 0.19\%$ | $\mathbf{30.20} \pm 0.609$ | $90.53\% \pm 0.23\%$ | $\mathbf{8.52} \pm 0.175$ |

Table D.12: Coverage and average set size results for ImageNet with RSCP with standard deviation.

## D.6 Extended study on the impact of $N_{\mathrm{MC}}$

| $N_{MC}$ | 128 | 256 | 512 | 1024 |
|---|---|---|---|---|
| CIFAR100 | 39.95 | 26.06 | 13.60 | **12.43** |
| ImageNet | 1000 | 59.12 | 19.68 | **10.34** |

Table D.13: **Average size vs. Number of Monte Carlo samples** $N_{MC}$**.** The experiment is conducted with PTT method. The base score is HPS. It could be seen that by increasing the number of Monte Carlo examples, we could further improve the efficiency of `RSCP+`, at the cost of higher computational expense.

In Sec. 5, we study the impact of the number of Monte Carlo samples $N_{\mathrm{MC}}$ on `RSCP+` on the CIFAR10 dataset. In this section, we extend the study to CIFAR100 and ImageNet. As shown in Tab. D.13, the average size could be reduced by increasing the number of Monte Carlo examples.

## D.7 IMPACT OF $T$

In this section, we conduct an empirical study on the impact of hyperparameter $T$ in PTT. We choose the CIFAR10 dataset with HPS as the base score. The results are shown in Fig. D.2. It could be seen that when choosing $T$ large enough, our method is not sensitive to the choice of $T$.

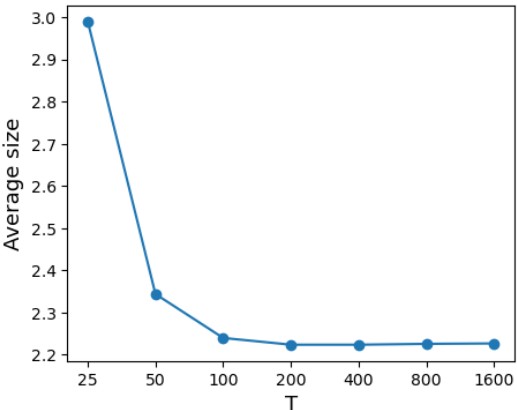

Figure D.2: Average size of prediction set vs. different $T$.

### D.8 COMPUTATIONAL OVERHEAD OF OUR METHODS

In this section, we discuss the computational overhead of our methods, PTT and RCT. The overhead could be divided into two parts: training cost and test-time cost.

**Training cost.** Our PTT method is training-free, hence it doesn't introduce any cost for training. For RCT, we compare the training time of RCT and ConfTr (Stutz et al., 2021) on the CIFAR10 dataset. For both of the methods, the last linear layer is finetuned for $N_{\text{epoch}} = 150$ epochs on 2 NVIDIA V100 GPUs. RCT takes around 95 minutes while ConfTr takes around 50 minutes. We argue that this computational overhead stems from randomized smoothing: training base model for randomized smoothing requires sampling a batch of examples for each training example (Cohen et al., 2019; Salman et al., 2019; Zhai et al., 2020), which significantly increases the computational expanse.

**Test-time cost.** During test time, our RCT method is equivalent to vanilla RSCP (Gendler et al., 2021). No additional expense is needed. For PTT, since a transformation is applied on base score $S$ at test time, there would be a computational overhead: see Fig. D.3. Diving deeper into each step,

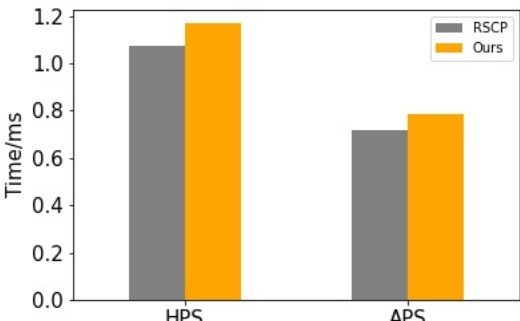

Figure D.3: Score computation time for each test example on CIFAR10 dataset. Comparing vanilla RSCP (RSCP) and RSCP with our training-free transformation (ours), it could be seen that the computational overhead is about 10%. Note that our robust conformal training method does not introduce additional computational cost at the test stage. The experiment is performed on 2 NVIDIA V100 GPU.

we could see that the computational cost for each example is $\Theta(N_{\text{MC}} K \log|D_{holdout}|)$ for ranking transformation and $\Theta(N_{\text{MC}} K)$ for sigmoid transformation. Therefore, the overall cost would be $\Theta(N_{\text{MC}} K \log|D_{holdout}|)$. We emphasize that this cost does not scale with the cost of the base model, which is desirable because it enables users to employ arbitrarily large base models without increasing the computational overhead of PTT.

# E    OTHER RELATED WORKS IN ROBUST CONFORMAL PREDICTION

Besides Gendler et al. (2021), there are other related works that studied robust conformal prediction (Einbinder et al., 2022a; Ghosh et al., 2023; Bastani et al., 2022). Bastani et al. (2022) focused on an online setting where the defender could interact with the attacker and collect information on adversarial distribution from history inputs, which is different from our setting which assumes the defender does not have information on the distribution of adversarial noises, except the assumption that they have limited norm. Ghosh et al. (2023) studied probabilistically robust conformal prediction. Formally, they studied the following guarantee:

$$\mathbb{P}_{X,Y,\delta}\{y_{n+1} \in C(\tilde{x}_{n+1} = x_{n+1} + \delta)\} \geq 1 - \alpha. \tag{E.1}$$

As Ghosh et al. (2023) pointed out, this could be regarded as a relaxed version of the adversarial robust coverage guarantee defined in Eq. (2): probabilistically robust conformal prediction provides a guarantee for perturbation $\delta$ following a specific distribution, while our $\texttt{RSCP+}$ provide adversarial robustness guarantee for any perturbation that satisfies $\|\delta\|_2 \leq \epsilon$.

## F    TABLE OF SYMBOLS

| Symbol | Description |
|---|---|
| $C(x)$ | Prediction set of input $x$ |
| $(x_{n+1}, y_{n+1})$ | Test input and corresponding label |
| $P_{xy}$ | Joint distribution of input $x$ and label $y$ |
| $\tilde{x}_{n+1}$ | Perturbed test input |
| $1 - \alpha$ | Target level of coverage |
| $\epsilon$ | Magnitude of adversarial noise |
| $C_\epsilon(x)$ | RSCP prediction set with noise level $\epsilon$ |
| $D_{\text{train}}$ | Training dataset |
| $D_{\text{cal}}$ | Calibration dataset |
| $\hat{\pi}$ | Probability output of classifier |
| $S$ | (Non-)conformity score function |
| $\tilde{S}$ | Randomized smoothed score defined by RSCP (Gendler et al., 2021) |
| $\tau$ | Threshold of vanilla conformal prediction |
| $\tau_{\text{adj}}$ | Adjusted threshold introduced by RSCP (Gendler et al., 2021) |
| $\Phi$ | Cumulative density function of standard Gaussian distribution |
| $S_{\text{RS}}$ | Randomized smoothed score defined in Eq. (9) |
| $\hat{S}_{\text{RS}}$ | Monte Carlo estimator for $S_{\text{RS}}$ |
| $\sigma$ | Smoothing strength of randomized smoothing |
| $Q_p(H)$ | The $p(1 + 1/|H|)$-empirical quantile of a set $H$ |
| $C_\epsilon^+(x)$ | Prediction set of RSCP+ with noise level $\epsilon$ |
| $\alpha_{\text{gap}}$ | Coverage gap introduced by RSCP (Gendler et al., 2021) |
| $\Phi_{\tilde{S}}$ | Cumulative density function of $\tilde{S}$ |
| $\mathcal{Q}$ | Transformation on base conformity score |
| $N_{\text{train}}$ | Number of Monte-Carlo examples used in RCT |
| $N_{\text{MC}}$ | Number of Monte-Carlo examples in calibration and test stages |
| $N_{\text{split}}$ | Number of random calibration-test split in evaluation |
| $b, T$ | Hyper-parameters of Sigmoid transformation |

Table F.1: Symbol table

