# OpenReview forum: "Provably Robust Conformal Prediction with Improved Efficiency"
_ICLR.cc/2024/Conference — ICLR 2024 poster_

### Official Review · Reviewer_y2Uk · 2023-10-26

**Soundness:** 3 good
**Presentation:** 2 fair
**Contribution:** 3 good
**Rating:** 6
**Confidence:** 3

**Summary:**

This paper proposes three main improvements over RSCP. It first built upon RSCP and use a high probability bound for the smoothed score approximation, in order to provide a more rigorous coverage guarantee in the adversarial setting. Then, it provides two tricks to improve the efficiency of the prediction sets. The first involves a general pipeline to modify the conformity score. The second is re-training the base model by mimicking the conformal prediction steps. The results suggest that the final method works better than the RSCP baseline.

**Strengths:**

1. The problem (first identified in the RSCP paper itself) about the missing step of high-probability bound is real, and this paper provides a good solution.
2. The PTT steps are quite interesting and could work with any nonconformity score (although it's not clear whether the efficiency is always improved).

**Weaknesses:**

1. It seems like RCT is not making too much difference, and it requires retraining the model. While it is an interesting idea, I wouldn't say the experiment supports that this helps, and is kind of distracting from the main idea of the paper.

2. It is unclear how to select $T$ (and to some extend $\sigma$).

3. Missing discussion of potential failure mode (see Q1)

**Questions:**

1. Is the intuition behind slope reduction that we "zoom in" to the nonconformity score around its 1-\alpha quantile? My intuition is that this essentially
	a. Section D.2 seems to be constantly flipping/inconsistent between maximizing/minimizing - are these mistakes?
	b. While the coverage guarantee always holds, are there cases where the PTT transform hurts the efficiency? For example, it seems like the optimality proof hinges on the unfiorm distribution, and if the noise is adversarial to this sigmoid transformation can PTT actually inflate more? It's just a bit difficult to imagine a transformation is dominating the original score.
2. How do we choose $T$ in practice? Obviously we have to fixed it ex-ante to avoid breaking the i.i.d. condition? In fact, given the proof/theorem, what prevents us from using $T=\infty$?

---

> ### Author Response · Authors · 2023-11-19
> **Response to Reviewer y2Uk (1/2)**
>
> Dear Reviewer y2Uk,
>
> Thank you for your positive feedback! We are glad to hear that you think that the problem about the missing step of high-probability bound is real, this paper provides a good solution, and the PTT steps are quite interesting and could work with any nonconformity score.
>
> Below we would like to address your questions and concerns in the weakness part with a detailed response.
>
> ---
>
> **Q1:** *Is the intuition behind slope reduction that we "zoom in" to the nonconformity score around its 1-\alpha quantile?*
>
> **A1:** Yes, our intuition is quite similar to what you describe! As we discussed in Sec. 4, conservativeness stems from the threshold inflation of RSCP/RSCP+, and we aim to alleviate this conservativeness by reducing the impact of threshold inflation. To achieve this, we give an analysis in Sec. 4.1 (see Eqn (19)-(22) in our draft) by local linear approximation (“zoom in”) which suggests we should reduce the slope at the threshold ($1-\alpha quantile). Intuitively, having a smaller slope means that the cdf is less sensitive to change in threshold.
>
> &nbsp;
>
> ---
>
> **Q1(a):** *Section D.2 seems to be constantly flipping/inconsistent between maximizing/minimizing - are these mistakes?*
>
> **A1(a):** Thank you for pointing this out! We realized there is a typo in the theorem statement:
> In our previous draft (Theorem D.11), we said “our transformed score could achieve the *largest* derivative” and it should be fixed as “smallest”. Similarly, in the statement we said “$\Phi^{\prime}_{\tilde S}(\tau)$ is *maximized* by” and it should be fixed as “minimized”. We have fixed all above typos in the revised draft – our goal is to minimize the slope, as discussed in Sec. 4.1. By Theorem D.11, we would like to show that the optimal distribution (having the smallest slope at the threshold) could be achieved by our transformation.
>
> &nbsp;
>
> ---
>
> **Q1(b):** *While the coverage guarantee always holds, are there cases where the PTT transform hurts the efficiency? For example, it seems like the optimality proof hinges on the unfiorm distribution, and if the noise is adversarial to this sigmoid transformation can PTT actually inflate more? It's just a bit difficult to imagine a transformation is dominating the original score.*
>
> **A1(b):** We would like to clarify that our transformation is not designed to “dominate” the original score , but to enhance its robustness to threshold inflation which is required in RSCP/RSCP+. In Appendix L (p.34 of our draft), we discussed this and showed that by our design, the transformed score will be **equivalent** to the original score under vanilla conformal prediction. For RSCP/RSCP+, however, the transformed score would be better against adversarial perturbation due to enhanced robustness.
>
> &nbsp;
>
> ---
>
> **Q2:** *How do we choose T in practice? Obviously we have to fixed it ex-ante to avoid breaking the i.i.d. condition? In fact, given the proof/theorem, what prevents us from using $T = \infty$?*
>
> **A2:** In Appendix F, we conduct a theoretical analysis of PTT and suggest we should choose T high enough. We realized there was a typo in the sentence of original Appendix F after Eq (F.3) and we have fixed it: we change it to “Hence with T large enough, we could achieve our goal…”. We are sorry for any confusion on this.
>
> We also add an empirical study on the impact of T on CIFAR10 in below Table R1 and summarize it in Appendix N.4 in the revised draft. The results suggest that our PTT method is not sensitive to the choice of T, as long as it’s sufficiently large. Hence, we simply choose $T=400$. We hypothesize that the reason the performance saturates after T becomes sufficiently large is that the linear approximation we performed in Sec 4.1 becomes less accurate: the linear term may not be dominant when the slope is small.
>
> | $T$       | 25   | 50    | 100   | 200   | 400   | 800   | 1600  |
> |-----------|------|-------|-------|-------|-------|-------|-------|
> | Avg. Size | 2.99 | 2.344 | 2.240 | 2.224 | 2.224 | 2.226 | 2.227 |
> Table R1: Impact of $T$ to prediction sets size on CIFAR10.

---

> ### Author Response · Authors · 2023-11-19
> **Response to Reviewer y2Uk (2/2)**
>
> **Weakness #1:** *It seems like RCT is not making too much difference, and it requires retraining the model. While it is an interesting idea, I wouldn't say the experiment supports that this helps, and is kind of distracting from the main idea of the paper.*
>
> **Ans:** Thank you for the comments! We partially agree with your comments that the RCT alone is more expensive and makes less improvement compared to PTT. However, as evident in Table 1 on p.9 of our draft, if we combine RCT with PTT, in some cases we can further improve the result of PTT, which shows the value of RCT.
>
> We also want to point out that: RCT does not need to train the model from scratch. As discussed in Appendix I.1, in RCT we froze the backbone and only finetuned the last linear layer. This approach could be much cheaper, compared with training from scratch. Therefore, we believe RCT will be useful for users who want to further boost performance and are less sensitive to computational cost.
>
> &nbsp;
>
> ---
>
> **Weakness #2:** *It is unclear how to select $T$ (and to some extend $\sigma$).*
>
> **Ans:** In the above Q2 response, we discussed the selection of $T$ and our additional experiment in Table R1 that our result is not sensitive to $T$. For $\sigma$, we followed the recommendation of [1] to use $\sigma = 2\epsilon$. We also conduct an empirical study on the impact of $\sigma$ in Appendix I.3 (Table I.6, p.30), which shows that this choice works well empirically.
>
> [1] Gendler et al. Adversarially robust conformal prediction. ICLR 2022.
>
> &nbsp;
>
> ---
>
> **Weakness #3:** *Missing discussion of potential failure mode.*
>
> **Ans:** As we discussed in the response to Q1(b), our transformed score is equivalent to the original score in vanilla conformal prediction, hence there would not be a failure mode under this setting. For robust conformal prediction, despite theoretical possibility, our empirical study does not observe a case where PTT inflates more than the original score. We will leave it as  future works to show theoretical possibility or impossibility results for such cases and explore the failure mode of the PTT method.
>
> &nbsp;
>
> ---
>
> **Summary**
>
> To summarize, we have:
> * Discussed the intuition behind our PTT method in **Q1**.
> * Clarified some typo fixes in **Q1(a)**.
> * Clarified some misunderstandings that our transformed score dominates the original score in **Q1(b)**.
> * Discussed the choice of hyperparameter T in **Q2**.
> * Clarified some advantages of our RCT method in **Weakness #1**.
> * Discussed the choice of hyperparameter T and $\sigma$ in **Weakness #2**.
> * Discussed the failure mode of PTT in **Weakness #3**.
>
> Please feel free to let us know if you have any additional comments and/or questions. We would be happy to discuss further!

---

> > ### Comment · Reviewer_y2Uk · 2023-11-21
> >
> > Thank you for the explanations. However I still feel my Q1b is not quite answered - while it is possible that future research could identify the "failure mode", I think the authors probably have a better idea as they designed this method. Again, by "failure", I don't mean when the proposed method will become invalid. Rather, I'm just curious about what might be a case when the transformation actually hurts efficiency. For example, why does zoom-in help? Does whether the $\approx$ in Eq(22) overestimate or underestimate imply lower/higher efficiency? Is there a way to design a simulated distribution of original nonconformity score such that PTT will actually hurt the efficiency? I think discussing such a failure mode does not hurt this paper  but actually helps this paper: the idea of doing such transformation is already novel and valuable, but such discussion will further gives us a better idea of when to apply this method and could potentially inspire more future research as well.

---

> ### Author Response · Authors · 2023-11-22
> **Author response to the rebuttal feedback from Reviewer y2Uk**
>
> Dear Reviewer y2Uk,
>
> Thank you for the suggestion! Despite that we did not find failure cases empirically, we agree that exploring potential failure mode will be theoretically interesting.
>
> Hence, we have revisited our 1D synthetic data example in Appendix D of our original draft, where the theoretical analysis is more feasible. Following your suggestion, we found that indeed under certain conditions, it is possible to create an example where our PTT method could generate a larger set than the original score. The key idea of the construction is to build a “slightly wrong” base score and carefully choose $\alpha$ and $\sigma$ to make PTT amplify this error.
>
> Our analysis is detailed and added to the Appendix N.6 of the revised draft and our result is in the Table R2 below. It can be seen that if we choose $\sigma = 0.3$, then the average size of PTT is slightly larger than the average size of the baseline (the original score without transformation).
>
> | $\sigma$ | Avg. Size(Baseline) | Avg. Size(PTT) |
> |--------|-----------------|----------------|
> | 0.2    |           0.529 |          0.511 |
> | 0.3    |           0.525 |          0.529 |
> Tab. R2: Average size for original score (Baseline) and PTT in our constructed failure case. We use $\alpha = 0.5$ and $\frac{\epsilon}{\sigma} = 0.01$.
>
>
> However, despite being theoretically possible, we also want to point out that this failure mode requires (1) a deliberately designed base score and (2) a very specific choice of $1-\alpha$ and $\sigma$ (Note that in this case, we choose $1- \alpha=0.5$, which is usually much lower than the setting in practice where high coverage is usually desired).  Hence, the chance that this kind of failure happens in practice is very rare – We didn’t observe this failure mode in our experiments, and we found that PTT works very well with improved efficiency across standard image classification benchmarks including cifar 10, cifar 100 and imagenet.
>
> Nevertheless, we think it is important to explore the potential failure modes of PTT and we sincerely thank the reviewer for this valuable suggestion and insights for us to consolidate our result and improve the draft! Accordingly, we have also added a sentence at the end of Sec 4.1 PTT to point out this theoretical possibility.
>
> We hope that this example has addressed your concern on the potential failure modes of the PTT method in **Q1(b)**. Please let us know if you still have any reservations, and we would be happy to discuss further!

---

> > ### Comment · Reviewer_y2Uk · 2023-11-23
> >
> > Thank you for adding this experiment - this is what I had in mind and addresses my concerns.

---

> > > ### Author Response · Authors · 2023-11-23
> > > **Thank you!**
> > >
> > > Dear Reviewer y2Uk,
> > >
> > > We are glad to hear that your concerns have been addressed. Thank you for the valuable feedback and helping us improving the draft!

---

### Official Review · Reviewer_SvdF · 2023-10-31

**Soundness:** 3 good
**Presentation:** 3 good
**Contribution:** 3 good
**Rating:** 8
**Confidence:** 4

**Summary:**

This paper studies the robust conformal prediction with the existence of adversarial noise. The authors first point out that the existing method **RSCP** has two major limitations in practice, and then propose a new framework named **RSCP+** to address two issues. To further reduce the size of the prediction set, this paper also develops two methods, PTT and RCT. Plenty of experiments are conducted to verify the effectiveness of RSCP+.

**Strengths:**

1. The background of this problem is well illustrated.

2. The framework of RSCP+ is more efficient compared with the existing RSCP. The novelty of this method is also significant.

3. Two specific methods PTT and RCT make this new framework more practical.

**Weaknesses:**

Overall, I think this submission is a good paper, but I have the following concerns.

**1. The literature review is not sufficient and complete.**

For the conformal prediction with adversarial noise, this author discussed only one related work Gendler et al. (2021). However, I find that a published work [1] is closely related to this problem. The authors should clarify the differences between the submission and [1]. In addition, [2] and [3] are also related works.

**2. The difference between RSCP+ and RSCP should be discussed earlier.**

As the most important contribution of this paper, the algorithmic design of RSCP+ appears until Page 4. It would be better to summarize the main idea or unique part of RSCP+ in the Introduction.

**3. The empirical comparison to [1] and [2] should be added to the experiments.**

[1] Ghosh, Subhankar, Yuanjie Shi, Taha Belkhouja, Yan Yan, Jana Doppa, and Brian Jones. "Probabilistically robust conformal prediction." In Uncertainty in Artificial Intelligence, pp. 681-690. PMLR, 2023.

[2] Bastani, Osbert, Varun Gupta, Christopher Jung, Georgy Noarov, Ramya Ramalingam, and Aaron Roth. "Practical adversarial multivalid conformal prediction." Advances in Neural Information Processing Systems 35 (2022): 29362-29373.

[3] Einbinder, Bat-Sheva, Stephen Bates, Anastasios N. Angelopoulos, Asaf Gendler, and Yaniv Romano. "Conformal Prediction is Robust to Label Noise." arXiv preprint arXiv:2209.14295 (2022).

**Questions:**

1. Is the ranking-transformation technique proposed in your paper or already proposed in previous work? If the latter, the citation should be added.

2. Why does the Sigmoid transformation require uniformly distributed scores?

3. For the ranking transformation, the new score $Q_{rank}\circ S_i$ is uniformly distributed when we consider the randomness from both $S_i$ and the hold-out set. In this vein, the new calibration scores $Q_{rank}\circ S_{i}: i=1,...,n$ and the test score $Q_{rank}\circ S_{n+1}$ are no longer independent. How can we guarantee the validity of RSCP+?

4. Is there any particular reason why the regression problem is not considered?

---

> ### Author Response · Authors · 2023-11-19
> **Response to Reviewer SvdF (1/2)**
>
> Dear Reviewer SvdF,
>
> Thank you for your positive feedback! We are glad to hear that you think that the background of the problem is well illustrated, the novelty of our method RSCP+ is significant, and our proposed PTT/RCT makes RSCP+ more practical.
>
> Below we would like to address your questions and concerns in the weakness part with a detailed response.
>
> ---
>
> **Q1:** *Is the ranking-transformation technique proposed in your paper or already proposed in previous work? If the latter, the citation should be added.*
>
> **A1:** Thank you for the questions! We have discussed in Appendix M (Theorem M.13) that calculating the rank is an elementary operation in the statistics community and is well-studied, where we have cited the work [Kuchibhotla (2020)]. We will add a sentence in Sec 4.1 to make it more clear. As far as we know, within the scope of robust conformal prediction, no previous work uses this transformation to improve efficiency.
>
> [Kuchibhotla (2020)] Arun Kumar Kuchibhotla. Exchangeability, conformal prediction, and rank tests, 2020.
>
> &nbsp;
>
> ---
>
> **Q2:** *Why does the Sigmoid transformation require uniformly distributed scores?*
>
> **A2:** We believe that there may be a slight misunderstanding here. In PTT, the Sigmoid transformation does not require uniformly distributed scores. As we discussed in Sec. 4.1, the idea is to turn the score distribution into a known distribution, so that we don’t need to craft a specific transformation for every new dataset. We choose a uniform distribution for two reasons:
> It’s easy to generate with our ranking transformation.
> It’s simple and has good theoretical properties, facilitating our analysis.
>
> &nbsp;
>
> ---
>
> **Q3:** *For the ranking transformation, the new score $Q_{rank} \circ S_i$ is uniformly distributed when we consider the randomness from both $S_i$ and the hold-out set. In this vein, the new calibration scores $Q_{rank} \circ S_i: i=1,\cdots, n$ and the test score $Q_{rank} \circ S_{n+1}$ are no longer independent. How can we guarantee the validity of RSCP+?*
>
> **A3:** This is a very good question, thank you for asking! We try to formalize your question as: $Q_{rank} \circ S_i$ could be viewed as a function of $S_i$ and the holdout set $D_{holdout}$: $Q_{rank} \circ S_i = f(S_i, D_{holdout})$. Hence, the scores are no longer independent because they depend on the same holdout set.
>
> However, conformal prediction only requires scores to be exchangeable [Kuchibhotla (2020)] (i.e. Any permutation on these variables does not change the joint distribution), which is satisfied by our transformed scores (because the scores are symmetric). We present a formal proof in Appendix N.1 in the revised draft.
>
> [Kuchibhotla (2020)] Arun Kumar Kuchibhotla. Exchangeability, conformal prediction, and rank tests, 2020.
>
> &nbsp;
>
> ---
>
> **Q4:** *Is there any particular reason why the regression problem is not considered?*
>
> **A4:** In the literature on adversarial robustness in the vision domain, the most popular benchmark is image classification, hence we follow RSCP paper to focus on classification problems. Nevertheless, we think this will be an interesting direction to explore and it is possible to extend our approach to the regression problem because the coverage idea is similar. We will leave it as an interesting future direction and discuss it in the future work paragraph.
>
> &nbsp;
>
> ---
>
> **Weakness #1:** The literature review is not sufficient and complete.*
>
> **Ans:** Thank you for pointing this out! Following your suggestion, we have included the works [1, 2] to the reference. We believe that our RSCP+ method is not directly comparable to the methods in [1 ,2] because the problem setting of [1, 2] is very different from ours as explained in below:
> * [1] studied probabilistic robust coverage (Definition 3 in [1]), which only works for some specific noise distribution and is a relaxed version of adversarial robust coverage studied in our paper.
> * [2] studied an online setting where the defender interacts with the adversary. Each round the defender could utilize the information on adversarial distribution collected in history. In our setting, the defender only assumes adversarial noise has limited norms without any further information. Hence, the task is more challenging.
>
> We have added a discussion paragraph in the revised draft to discuss the difference between our work and [1, 2] in Appendix N.2.
>
> &nbsp;
>
> ---
>
> **Weakness #2:** *The difference between RSCP+ and RSCP should be discussed earlier.*
>
> **Ans:** Thank you for the suggestions. Following your suggestion, we have added a summary of the main idea of RSCP+ in the revised draft.

---

> ### Author Response · Authors · 2023-11-19
> **Response to Reviewer SvdF (2/2)**
>
> **Weakness #3:** *The empirical comparison to [1] and [2] should be added to the experiments.*
>
> **Ans:** Thank you for your advice! Following your suggestions, we studied [1][2] and discussed their differences with our work. We found that the empirical results of [1][2] may not be directly comparable to ours: as we discussed in Weakness #1, both works chose a different setting than ours. As far as we know, our RSCP+ framework is the first to provide a valid adversarial robustness guarantee for conformal prediction.
>
> [1] Ghosh et al. "Probabilistically robust conformal prediction." UAI 2023.
>
> [2] Bastani et al. "Practical adversarial multivalid conformal prediction." NeurIPS 2022.
>
> &nbsp;
>
> ---
>
> **Summary**
>
> To summarize, we have:
> * Discussed the existence of ranking transformation in previous works in **Q1**.
> * Clarified the reason we construct a uniform-distributed score in **Q2**.
> * Explained the validity of our ranking transformation in **Q3**.
> * Discussed why we focus on classification problems in **Q4**.
> * Discussed related works mentioned by the reviewer and compared them with our work in **Weakness #1**.
> * Applied a modification recommended by the reviewer in **Weakness #2**.
> * Discussed the difference between our work and [1, 2] in **Weakness #3**.
>
> Please feel free to let us know if you have any additional comments and/or questions. We would be happy to discuss further!

---

> > ### Comment · Reviewer_SvdF · 2023-11-22
> >
> > I appreciate the detailed responses from the authors. All my concerns have been addressed. I have raised my score.

---

> > > ### Author Response · Authors · 2023-11-22
> > > **Thank you**
> > >
> > > Thank you for the reply! We are glad to hear that all your concerns have been well addressed by our rebuttal response, and we are thankful for raising the score. We would like to thank you and all the other reviewers for the valuable feedback to help us improve the draft!

---

### Official Review · Reviewer_mH5T · 2023-10-31

**Soundness:** 3 good
**Presentation:** 2 fair
**Contribution:** 3 good
**Rating:** 8
**Confidence:** 3

**Summary:**

This paper tackles the problem of robust conformal prediction under adversarial perturbations. The authors study the RSCP framework, a combination of randomized smoothing and conformal prediction, proposed previously, and 1) identify a technical flaw in RSCP's certification guarantee due to the Monte Carlo estimation of the non-conformity score. 2) propose a modified RSCP (RSCP+) framework that circumvents this issue to provide a solid certificate in practice, and 3) propose two new techniques for improving the efficiency of the prediction sets based. The first one (PTT) is a training-free method for computing the non-conformity scores, based on a simple yet effective transformation. The second (RCT) is a training-based approach to improve the robust conformal training of the base classifier itself. Combining PTT+RCT, the authors evaluate their new RSCP+ framework on a variety of network architectures and datasets, showcasing their effectiveness.

**Strengths:**

- The paper tackles an important subject: uncertainty quantification via conformal prediction in the presence of adversarial perturbations
- The paper is well written for the most part (see Weaknesses for some comments), and the authors go into great lengths to provide details, as seen by the dense Appendix. Technical contributions look sound to me.
- The paper correctly identifies a technical flaw in a prior framework (RSCP), and presents a correction based on the Hoeffding bound to establish a sound framework: RSCP+
- The proposed efficiency enhancing techniques (PTT and RCT) seem to perform well in practice.
- The authors verify their results on three vision benchmarks: CIFAR10, CIFAR100, and ImageNet

**Weaknesses:**

- The paper, to its credit, tackles three different techniques, which has unfortunately degraded the reading experience. The proposed techniques have little to do with each other, and jumping between them was a bit hard to grasp in the first few reads. Having to fit all this in the page limit certainly does not help the authors. There is also an over reliance on the Appendix, which made the reading experience very choppy. I am not sure what is the best way to tackle this frankly.

- One thing I found missing was that, if the original RSCP framework is flawed (due to practical limitations of having to rely on Monte Carlo sampling to estimate the non-conformity scores), then why does it still work well in practice? The explanation in Appendix A makes it seem like the method will yield catastrophic results, but that does not seem to be the case. It would greatly improve the motivation for this work if this part is properly explained, preferably in the main text, as it is key to the main motivation for this work.

**Questions:**

- The PTT method requires another set of held-out, independent of the calibration set. In order for a fair comparison with vanilla RSCP+, would it make sense to have the combined holdout and calibration sets as the baseline calibration set for vanilla RSCP+? This would eliminate any notion of PTT improvement coming from having access to more data, rather than the transformation.

- How were the adversarial examples constructed? is it vanilla PGD? or similar to the SmoothAdv [Salman et al., 2019] paper?

- How well does the RSCP+ framework perform on unperturbed data? are the prediction sets also trivial for clean data?

---

> ### Author Response · Authors · 2023-11-19
> **Response to Reviewer mH5T (1/2)**
>
> Dear Reviewer mH5T,
>
> Thank you for the positive feedback! We are glad to hear that you think our paper tackles an important subject, well written for most part and the technical contributions look sound to you!
>
> Please see our answers to your questions and we would love to address the weakness part.
>
> ---
>
> **Q1:** *In order for a fair comparison with vanilla RSCP+, would it make sense to have the combined holdout and calibration sets as the baseline calibration set for vanilla RSCP+?*
>
> **A1:** Thank you for the suggestions! Following your suggestion, we conducted additional experiments for RSCP+ using the combination of the holdout set and calibration set as the baseline calibration set. We find that the results are the same, as shown in Table N.2: they still produce trivial results, where the average prediction set size is the same as the number of classes.
>
> &nbsp;
>
> ---
>
> **Q2:** *How were the adversarial examples constructed? is it vanilla PGD? or similar to the SmoothAdv [Salman et al., 2019] paper?*
>
> **A2:** Thanks for the question! We discussed in the Appendix I.2 (p.28) in the main draft that we follow the approach of RSCP in [1, Sec 5.1, Attack algorithm], which uses SmoothAdv [Salman et al., 2019] to generate adversarial examples. We optimize objective Eq. (S) in Sec 2.2 in [Salman et al., 2019] with PGD and the gradient approximation in [Salman et al., 2019].
>
> [1] Gendler etal. Adversarially robust conformal prediction. ICLR 2022.
> [Salman et al., 2019] Provably Robust Deep Learning via Adversarially Trained Smoothed Classifiers. NeurIPS 2019
>
> &nbsp;
>
> ---
>
> **Q3:** *How well does the RSCP+ framework perform on unperturbed data? are the prediction sets also trivial for clean data?*
>
> **A3:** The RSCP+ alone (w/o PTT or RCT) generates a trivial prediction set for **both** clean data and perturbed data. In other words, for Cifar 10 and Cifar 100, RSCP+ alone will give trivial prediction set sizes of 10 and 100, for both settings. As we discussed in Sec. 4, the key reason for trivial prediction is conservativeness, motivating us to design PTT/RCT to address this problem.
>
> &nbsp;
>
> ---
>
> **Weakness #1:** *The paper, to its credit, tackles three different techniques, which has unfortunately degraded the reading experience. The proposed techniques have little to do with each other, and jumping between them was a bit hard to grasp in the first few reads. Having to fit all this in the page limit certainly does not help the authors. There is also an over reliance on the Appendix, which made the reading experience very choppy. I am not sure what is the best way to tackle this frankly.*
>
> **Ans:** Thank you for sharing your thoughts! Please let us clarify below. The key goal of our paper is to build a prediction set with a certified robustness guarantee. For this goal, we first proposed RSCP+ in Sec 3 to amend the theoretical flaw in previous works, and then proposed PTT/RCT in Sec 4 to address additional challenges to improve the efficiency of robust conformal prediction. We will make it more clear in the motivation and introduction section to show that these three parts are related and necessary for our goal of developing robust conformal prediction with guarantee:
> RSCP+ (obtain certified robustness guarantee)
> PTT and RCT (improve the efficiency of RSCP+).
>
> We plan to organize the Appendix into three main parts: supplementary contents related to RSCP+, supplementary contents related to PTT, and supplementary contents related to RCT. Note that this is not revised yet in the current revised draft, as we are referring to many different appendix in the original draft, so we think it will be less confusing to keep the original appendix for now. But we plan to organize the Appendix as above in the camera-ready version, and we hope that this will make the supplementary materials more clear and organized to improve the readability. Thank you for helping us improve the draft!
>
> &nbsp;
>
> ---
>
> **Weakness #2(a):** *One thing I found missing was that, if the original RSCP framework is flawed (due to practical limitations of having to rely on Monte Carlo sampling to estimate the non-conformity scores), then why does it still work well in practice?*
>
> **Ans:** As we discussed in Sec. 3, the flaw of RSCP is in its guarantee. The RSCP framework still works empirically because it’s very conservative (as reflected in the high coverage in Table I.3 in the main draft). However, the flaw in guarantee of [1] may give users a false sense of safety. The provable robustness guarantee is important – because without provable guarantees, there is always a possibility that the result will break with stronger attacks. This is the overarching idea of “robustness certification” in the adversarial robustness literature, where it is important to provide robustness “guarantees”, as an empirically robust method could fail at more advanced attacks in the future. Hence, we proposed RSCP+ to address this flaw and provide provable robustness guarantee.

---

> ### Author Response · Authors · 2023-11-19
> **Response to Reviewer mH5T (2/2)**
>
> **Weakness #2(b):** *The explanation in Appendix A makes it seem like the method will yield catastrophic results, but that does not seem to be the case.*
>
> **Ans:** In Appendix A, we would like to show the theoretical difficulty in **directly applying** error bounds on RSCP.  As we discussed in Appendix A, the bound will be very loose, and yield trivial prediction sets if we directly use it to construct a prediction set. Actually, with our approach, we derive a tighter bound and thus get a practical and guaranteed framework, RSCP+.
>
> &nbsp;
>
> ---
>
> **Summary**
>
> To summarize, we have:
> * Updated empirical results for baseline with the recommended modifications in **Q1**.
> * Clarified the construction of adversarial examples in **Q2**.
> * Clarified the performance of RSCP+ alone on clean data in **Q3**.
> * Discussed the relation of our three techniques in  **Weakness #1**.
> * Discussed the reason why RSCP is flawed but still works empirically in **Weakness #2(a)**.
> * Clarified our explanation in Appendix A in **Weakness #2(b)**.
>
> Please feel free to let us know if you have any additional comments and/or questions. We would be happy to discuss further!

---

### Official Review · Reviewer_Tjzn · 2023-11-01

**Soundness:** 3 good
**Presentation:** 3 good
**Contribution:** 2 fair
**Rating:** 6
**Confidence:** 4

**Summary:**

This paper identifies two limitation of Randomized Smoothed Conformal Prediction (RSCP): (1) its robustness guarantee is questionable in practical applications; (2) it often yields large uncertainty sets. To address the first problem, the paper propose a novel framework called RSCP+ to provide provable robustness guarantee. For the second issue, the paper introduces two innovative techniques: Post-Training Transformation (PTT) and Robust Conformal Training (RCT), reducing prediction set size.

**Strengths:**

1. This approach is novel and theoretically sound for conformal prediction. This paper considers the conformal prediction in an adversarial setting where exchangeability is violated; thus traditional methods fail to guarantee coverage. The authors improves RSCP by adjusting its non-conformity score and bounds the estimation error.

2.  This approach is theoretically grounded and computationally efficient. This paper designs two post-training transformation functions: Ranking Transformation and Sigmoid Transformation, aim at minimizing $\alpha_{gap}$ by reducing the slope of $\Phi_{\tilde{S}}(\tau)$.

3. Inspired by conformal training, the authors propose robust conformal training to enhance efficiency at training stage. Empirical results show that this method is efficient.

**Weaknesses:**

1. RSCP+ fail to construct informative prediction sets independently: it tends to give the whole prediction sets (include all classes) as observed on Cifar-10, Cifar-100, ImageNet.

2. RSCP+ generates relatively large prediction sets on dataset such as ImageNet even with PTT, limiting its application.

3. The impact of number of Monte Carlo on RSCP+ remains ambiguous, since the experiment is only conducted on Cifar-10.

**Questions:**

1. The Theorem 1 in [R1] provides a guarantee for RSCP prediction set; moreover, I didn't find noticeable coverage violation in empirical results. Thus, I'm confused why you refine the non-conformity score to bound the estimation error?

2. Can you share the result of PTT+RCT of RSCP+ on ImageNet?

3. Can you conduct additional experiments on the impact of number of Monte Carlo on RSCP+?

4. Considering sigmoid transformation, how you determine the optimal value for $T$.

5. Although PTT and RCT successfully improve efficiency (as shown by empirical results), I don't understand your statement that the coverage gap $\alpha_{gap}$ reflects the conservativeness of RSCP. Can you provide more theoretical or empirical explanation?

6. In Section 4, the paper points out that RSCP and RSCP+ is conservative: RSCP gives a larger prediction set on both clean and perturbed data; this conservativeness then passes on to RSCP+, as supported by Table 1 and Table 2. However, the complementary results in Appendix I don't strongly indicates RSCP is conservative: for example, on Cifar-10, RSCP generates prediction sets of average size 2.751 based on APS. Is it improper to say that RSCP+ is conservative simply because it builds upon RSCP?

---

> ### Author Response · Authors · 2023-11-19
> **Author Response to Reviewer Tjzn (1/3)**
>
> Dear Reviewer Tjzn,
>
> Thank you for your constructive feedback! We are glad to hear that you think our approach is novel, theoretically sound, and computationally efficient!
>
> In the below response, we provide detailed answers and clarifications to address your questions and concerns.
>
> ---
> **Q1:** *The Theorem 1 in [R1] provides a guarantee for RSCP prediction set; moreover, I didn't find noticeable coverage violation in empirical results. Thus, I'm confused why you refine the non-conformity score to bound the estimation error?*
>
> **A1:** We didn’t see the reference provided for the [R1] you are referring to, but we believe you are referring to the RSCP paper by Gendler et al [1].
>
> The Theorem 1 in [1] is valid *theoretically* but becomes invalid in practice because it assumes the smoothed non-conformity score $\tilde{S}$ is used to construct the prediction set $C_{\delta}$ [1, their Eq (8) at p.4] (or see Eq (9) in Sec 2 in our main draft at p.3), while this is approximated by the Monte Carlo estimator $\hat S_{RS}$ (see Eq (12) in the main draft) in practice. This gap between theory and practice is usually filled by giving an estimation bound in randomized smoothing literature, while this bound is missing in [1] as we point out in Sec 3.
> Without giving an estimation error bound of $\hat S_{RS}$, the computed prediction set is invalid in [1], and hence the guarantee in Theorem 1 in [1] will not hold in experiments and in practice. This is why we said at the beginning of Sec 3 that there is a missing step (the step 2: estimation error bound in our main draft) in [1] and missing this step makes the RSCP’s guarantee invalid, despite working empirically.
>
> To fix this issue, we propose to incorporate the Monte Carlo estimator $\hat S_{RS}$ directly as the non-conformity score, as discussed in the first 3 paragraphs in Sec 3 in our draft (p.4). We show in Fig 2 and prove in our Theorem 1 and Corollary 2 to incorporate the estimation error for $\hat S_{RS}$ such that the prediction set $C_{\epsilon}^+$ satisfies robust coverage guarantees in Eq (2) with Monte Carlo estimation parameters (e.g. $b_{hoef}$).
>
>
> [1] Gendler et al. Adversarially robust conformal prediction. ICLR 2022.
>
>
> &nbsp;
>
> ---
> **Q2:** *Can you share the result of PTT+RCT of RSCP+ on ImageNet?*
>
> **A2:** Existing conformal training methods, e.g. [2][3] are not scalable to the ImageNet and focus on smaller datasets like CIFAR10 and CIFAR100. As our proposed robust conform training (RCT) builds upon existing conformal training methods (e.g. [2]) with additional modification to incorporate robustness (which is our novelty in sec 4.2), our RCT currently also suffers from the same issues as existing conformal training methods [2][3].
>
> Nevertheless, we would like to highlight that RCT is not the only contribution of our work. In fact, we have proposed another novel training-free method called PTT (see sec 4.1 in our draft) that is scalable to ImageNet. Indeed, for practitioners working on datasets at ImageNet scale, we recommend using our PTT method which is scalable and computationally efficient.
>
> In particular, we have reported in Table 2 of our main draft that the prediction set size could be reduced to **~59** for ImageNet, which gives meaningful and non-trivial predictions unlike baselines. Furthermore, the performance of PTT could be boosted (i.e. the average prediction size can be further reduced to a size around **10** for ImageNet, which is significant) by increasing the number of Monte-Carlo examples, as we discussed immediately in the response below (Q3 and Table R1). This suggests that PTT is very effective and scalable. The result is summarized in the revised draft in Appendix N.3.
>
> [2] Stutz et al. "Learning Optimal Conformal Classifiers." ICLR 2021.
>
> [3] Einbinder et al. "Training uncertainty-aware classifiers with conformalized deep learning." NeurIPS 2022.
>
>
> &nbsp;
>
> ---
> **Q3:** *Can you conduct additional experiments on the impact of number of Monte Carlo on RSCP+?*
>
> **A3:**
> Thank you for the suggestion! Following your suggestion, we have conducted additional experiments and reported the results below and the revised draft in Appendix N.3 (Table N.1). As shown in Appendix N.3,  the size of the prediction set could be further reduced by increasing the number of Monte Carlo examples: e.g. for ImageNet, we can further reduce the prediction size by 3-5.7 $\times$, which is a significant improvement.
>
> | $N_{MC}$   | 256    | 512   | 1024      |
> |----------|--------|-------|-----------|
> | CIFAR100 |  26.06 | 13.60 | **12.43** |
> | ImageNet | 59.12  | 19.68 | **10.34** |
>
> Table R1: Impact of $N_{MC}$ to prediction sets size on CIFAR100 and ImageNet.

---

> ### Author Response · Authors · 2023-11-19
> **Author Response to Reviewer Tjzn (2/3)**
>
> **Q4:** *Considering sigmoid transformation, how you determine the optimal value for T*
>
> **A4:** In Appendix F (p.25), we conduct a theoretical analysis of PTT and suggest that we should choose T high enough because it could reduce the slope near the threshold and thus reduce conservativeness (i.e. the derivative in Eq (F.3) will become close to zero if T is large).
>
> We realized there was a typo in the sentence of original Appendix F after Eq (F.3) and we have fixed it: we changed it to “Hence with T large enough, we could achieve our goal…”. Thank you for the question!
>
> We also conduct an additional experiment on the impact of T on CIFAR10 in below Table R2 and summarize it in Appendix N.4 in the revised draft. The results suggest that our PTT method is not sensitive to the choice of T, as long as it’s sufficiently large.
>
> | $T$       | 25   | 50    | 100   | 200   | 400   | 800   | 1600  |
> |--------|------|-------|-------|-------|-------|-------|-------|
> | Avg. Size | 2.990 | 2.344 | 2.240 | 2.224 | 2.224 | 2.226 | 2.227 |
> Table R2: Impact of $T$ to prediction sets size on CIFAR10.
>
>
> &nbsp;
>
> ---
>
> **Q5:** *Although PTT and RCT successfully improve efficiency (as shown by empirical results), I don't understand your statement that the coverage gap reflects the conservativeness of RSCP. Can you provide more theoretical or empirical explanations?*
>
> **A5:** Recall that the coverage gap is defined as the change in coverage after threshold inflation:  $\alpha_{\text{gap}} = \alpha - \alpha_{\text{adj}}$ (Eq. 19 in our draft), and $\textbf{conservativeness}$ is defined as the change in the prediction sets size after threshold inflation: $\textbf{conservativeness} =  \mathbb E (|C_\epsilon(x)| - |C(x)|)$(Eq. K.1 in our draft). Intuitively, given a non-conformity score function, higher coverage implies a larger prediction set.
>
> When the coverage gap goes to 0, the predictor converges to the vanilla conformal prediction and then the conservativeness is also 0.
> In Appendix D (p.19-22) we theoretically analyze a 1-D example and also show that in Table D.1 (p.20), $\alpha_{\text{gap}}$ is positively correlated with conservativeness, supporting our intuition.
>
> &nbsp;
>
> ---
> **Q6:** *However, the complementary results in Appendix I don't strongly indicates RSCP is conservative: for example, on Cifar-10, RSCP generates prediction sets of average size 2.751 based on APS. Is it improper to say that RSCP+ is conservative simply because it builds upon RSCP?*
>
> **A6:** In this paper, conservativeness is measured by the set inflation compared with *vanilla conformal prediction*. As shown in Figure I.1 (b) (p.29), the RSCP has a much larger average size (almost 2-2.5$\times$ larger) compared to vanilla conformal prediction, thus we say it’s conservative.
>
> In Sec. 4, we said RSCP+ inherits the conservativeness of RSCP to emphasize that RSCP+ is also conservative because it uses the threshold inflation similar to RSCP, which is a root cause of conservativeness. To reduce this conservativeness, we propose both PTT and RCT in sec 4.1 and sec 4.2 and show that we can effectively reduce the set size by 11.57%-31.12% in the Table I.3 that you are referring to.

---

> ### Author Response · Authors · 2023-11-19
> **Author Response to Reviewer Tjzn (3/3)**
>
> **Weakness #1**: *RSCP+ fail to construct informative prediction sets independently: it tends to give the whole prediction sets (include all classes) as observed on Cifar-10, Cifar-100, ImageNet.*
>
> **Ans:** Your understanding is correct and this is exactly the main reason that motivates us to propose PTT/RCT in Sec 4 to address this issue. With our proposed PTT and RCT, we can improve the prediction set size of RSCP+ significantly by 3.72-16.9$\times$, as reported in Table 1 and 2.
>
> &nbsp;
>
> ---
> **Weakness #2:** *RSCP+ generates relatively large prediction sets on dataset such as ImageNet even with PTT, limiting its application.*
>
> **Ans:**
> As far as we know, our PTT method provides **the first non-trivial result on ImageNet** as reported in Table 2 in the main draft (p.9). It can be seen in Table 2 that the baselines [1] gives trivial results (prediction set size = full class number = 1000) on ImageNet, while with our method, we can already reduce it by 10-16.9 $\times$ to a size of 59.12.
>
> In fact, as we discussed in the Q3 and Appendix N.3, following your suggestion, our results can be further improved by ~5$\times$ if we increase the number of monte-carlo samples $N_{MC}$, reaching a prediction set size equal to **10.34** for ImageNet. This is equal to an overall 96.7$\times$ improvement compared to the baseline, which is significant.
>
> This suggests that our proposed RSCP+ and PTT are scalable and can be used in real-world applications (e.g. safety-critical tasks) where guaranteed robustness is of significant importance.
>
>
> &nbsp;
>
> ---
> **Weakness #3:** *The impact of number of Monte Carlo on RSCP+ remains ambiguous, since the experiment is only conducted on Cifar-10.*
>
> **Ans:** Following your suggestion, we have conducted additional experiments on ImageNet and discussed in the above response for **Q3 (see Table R1)** and Appendix N.3 in the revised draft. We found that the size of the prediction set could be reduced further by 3-5.7 $\times$ if we increase the number of Monte Carlo examples.
>
>
> &nbsp;
>
> ---
> **Summary**
>
> To summarize, we have:
>
> * Discussed the motivation for our RSCP+ method in **Q1**.
>
> * Discussed the scalability of our methods in **Q2**.
>
> * Provided additional results on the impact of the number of Monte Carlo examples in **Q3**.
>
> * Discussed the choice of hyperparameter T in **Q4**.
>
> * Provided intuition and empirical evidence for the statement that the coverage gap reflects the conservativeness of RSCP in **Q5**.
>
> * Clarified why RSCP is conservative in **Q6**.
>
> * Clarified that our proposed PTT and RCT can significantly improve RSCP+ in **Weakness #1**.
>
> * Clarified RSCP+ is the first provable robust conformal prediction that scales to ImageNet in **Weakness #2**.
>
> * Provided additional experiments on Cifar 100 and ImageNet to address **Weakness #3**.
>
> We believe we have addressed all your concerns. Please let us know if you still have any reservations and we would be happy to discuss further!

---

> ### Author Response · Authors · 2023-11-20
> **Request for rebuttal feedback from Reviewer Tjzn**
>
> Dear Reviewer Tjzn,
>
> We believe that we have addressed all your concerns with additional new experiments (see General response and Appendix N). Please let us know if you still have any reservations and we would be happy to address them. Thank you!

---

> > ### Comment · Reviewer_Tjzn · 2023-11-21
> > **Thank you for the reseponse**
> >
> > Thank you for the detailed response. My concerns have been well addressed. So I am going to improve my score to 6.

---

> > > ### Author Response · Authors · 2023-11-21
> > > **Thank you**
> > >
> > > Dear Reviewer Tjzn,
> > >
> > > Thank you for the reply, we are glad to hear that all your concerns have been well addressed by our rebuttal response. We would like to thank you and all the other reviewers for the valuable feedback to help us improve the draft!

---

### Author Response · Authors · 2023-11-19
**General Response: Overview of new results**

In response to all reviewer comments, we have conducted several new experiments and made some changes to existing manuscript content. New text is written in brown color for visibility in the PDF.

# New experiments
We have compiled all the new results and discussion in Appendix N (p.36-p.37) in the PDF with a below summary:
* In Appendix N.3, we conduct an extended study on the impact of the number of Monte Carlo examples $N_{MC}$ on CIFAR100 and ImageNet. Results show that the size of the prediction set could be further reduced by increasing the number of Monte Carlo examples.
* In Appendix N.4, we studied the impact of hyperparameter T on CIFAR10. The results suggest that our PTT method is not sensitive to the choice of T, as long as it’s sufficiently large.
* In Appendix N.5, we address the concern that our PTT method uses more data by adding the holdout set to calibration data for the baseline, as recommended by Reviewer mH5T. The results are still trivial for the baseline, where the average prediction set size is the same as the number of classes.

# Other Changes
* We modified Sec.1 to make outline of the paper more clear, and discuss the difference between RSCP and RSCP+.
* In Appendix N.1, we presented a formal proof showing why our ranking transformation preserves exchangeability.
* In Appendix N.2, we discussed the difference between our paper and related works.
* Fixed typos.

---

### Meta-Review · Area_Chair_yMWr · 2023-12-11

**Metareview:**

Conformal prediction allows to construst set that contains labels of observed objets that is guaranteed to be valid with a high probability. It operates under the classical iid assumptions. It has been recently shown that the coverage guarantee can be corrupted by the presence of adversarial examples because it essentially violates the assumptions. A remedy was to leverage randomized smoothing in order to lower the influence of these examples on the loss of coverage. This paper highlight an important drawback of such approach. The core issue is that the Monte Carlo approximation error can invalidate the approach and also the size of the sets tend to be wide which is undesirable in practice. The contribution of the paper is to discuss such issues and provide *provably* robust conformal prediction sets under adversarial attacks by carefully taking into account the Monte Carlo error when smoothing the score functions.


> Minor comments:

- I would recommend moving the proof of Theorem 1 to the appendix to make it cleaner and less dense and also save space to cleanly define the method in the corollary 2. A pseudo code would be nice for example.

- The MC correction comes with an additional price in the coverage moving from $1 -\alpha$ to $1 - \alpha + 2 \beta$. What if I choose $\beta = \alpha / 2$? More discussions and illustrations on this would be nice. For instance I would expect a *decrease* in confidence to get instead $1 - \alpha - 2 \beta$. How $\beta$ is chosen wrt to $\alpha$? In the experiments, $\beta$ seems to be chosen independently of $\alpha$, which I find quite strange.

- If the previous method tends to be overly conservative as claimed by the authors, how come it can loose coverage in practice? This seems  quite counterintuitive. It should be nice to double check or illustrate more if RSCP+ is not (often) included in RSCP. In that case, the mentioned flaw seems quite minor. That being said, I agree that the certification is lost anyway and it is quite important to mention as done in this paper.

- The strategy to reduce the conformal set size induces an additional loss in the dataset. One needs another holdout independent data to make the transformation. The holdout data could be instead used to improve the efficiency of the classifier or have a larger calibration size. I find that this is not discussed enough. Furthermore, holdout set size impacts the proposed method?

**Justification For Why Not Higher Score:**

I believe that more discussions and illustration on the impact of adversarial example should be done. What if $\epsilon$ in $\|\tilde x_{n+1} - x_{n+1} \| \leq \epsilon$ is too large or even some others training data themselves are adversarial example? what if they are in the holdout data?

Impact of the hyper parameters etc ... Since the paper mentioned a potential flaw in previous approach, I would expect a safer approach here really discussing and crash-testing the proposed method.

**Justification For Why Not Lower Score:**

The overall impression is quite positive among the reviewers and I also believe that the paper propose a nice and simple approach to robustly conformal prediction methods.

---

### Decision · Program_Chairs · 2024-01-16

Accept (poster)